# Nuclear PTEN safeguards pre-mRNA splicing to link Golgi apparatus for its tumor suppressive role

Shao-Ming Shen[1], Yan Ji[2], Cheng Zhang[1], Shuang-Shu Dong[2], Shuo Yang[1], Zhong Xiong[1], Meng-Kai Ge[1], Yun Yu[1], Li Xia[1], Meng Guo[1], Jin-Ke Cheng[3], Jun-Ling Liu[1,3], Jian-Xiu Yu[1,3] & Guo-Qiang Chen[1]

Dysregulation of pre-mRNA alternative splicing (AS) is closely associated with cancers. However, the relationships between the AS and classic oncogenes/tumor suppressors are largely unknown. Here we show that the deletion of tumor suppressor PTEN alters pre-mRNA splicing in a phosphatase-independent manner, and identify 262 PTEN-regulated AS events in 293T cells by RNA sequencing, which are associated with significant worse outcome of cancer patients. Based on these findings, we report that nuclear PTEN interacts with the splicing machinery, spliceosome, to regulate its assembly and pre-mRNA splicing. We also identify a new exon 2b in GOLGA2 transcript and the exon exclusion contributes to PTEN knockdown-induced tumorigenesis by promoting dramatic Golgi extension and secretion, and PTEN depletion significantly sensitizes cancer cells to secretion inhibitors brefeldin A and golgicide A. Our results suggest that Golgi secretion inhibitors alone or in combination with PI3K/Akt kinase inhibitors may be therapeutically useful for PTEN-deficient cancers.

[1] Department of Pathophysiology, Key Laboratory of Cell Differentiation and Apoptosis of Chinese Ministry of Education, Shanghai Jiao Tong University School of Medicine (SJTU-SM), Shanghai 200025, China. [2] Institute of Health Sciences, Shanghai Institutes for Biological Sciences of Chinese Academy of Sciences and SJTU-SM, Shanghai 200025, China. [3] Department of Biochemistry and Molecular Cell Biology, Shanghai Key Laboratory of Tumor Microenvironment and Inflammation, SJTU-SM, Shanghai 200025, China. These authors contributed equally: Shao-Ming Shen, Yan Ji. Correspondence and requests for materials should be addressed to S.-M.S. (email: smshen01@sibs.ac.cn) or to G.-Q.C. (email: chengq@shsmu.edu.cn)

Gene expression in eukaryotes is finely controlled by complex regulatory processes that affect all steps of RNA expression. Inside these processes, one of the crucial steps is the constitutive splicing of pre-mRNA during which intronic sequences are removed and exonic sequences joined to form the mature messenger RNA (mRNA). Another regulation during this process is alternative splicing (AS), leading to the generation of several coding or non-coding mRNA variants from the same gene. Therefore, one of the main consequences of AS is to diversify the proteome through the synthesis of various protein isoforms displaying different biological activities[1]. The AS is tightly controlled across different tissues and developmental stages, and its dysregulation is closely associated with various human diseases including cancers. In the last decade, the development of high-throughput and systematic transcriptomic analyses together with the improvement of bioinformatic tools have extensively been increasing the amount of expression data regarding splice variants in cancers[1–3], and have revealed widespread alterations in AS relative to those in their normal tissue counterparts[4–7]. The existence of cancer-specific splicing patterns likely contributes to tumor progression through modulation of every aspect of cancer cell biology[8,9]. The identification of the AS isoforms expressed in tumors is therefore of utmost relevance to unravel novel oncogenic mechanisms and to develop new therapeutic strategies.

The splicing process is carried out by the spliceosome, a large complex of RNA and proteins consisting of five small nuclear ribonucleoprotein particles (snRNPs: U1, U2, U4, U5 and U6) and more than 200 ancillary proteins[10]. Each snRNP consists of a snRNA (or two in the case of U4/U6) and a variable number of complex-specific proteins. As well shown, AS is pathologically altered to promote the initiation and/or maintenance of cancers due to mutations in critical cancer-associated genes that affect splicing[5,6], and mutations or expression alterations of genes that affect components of the spliceosome complex[11–16]. It was also reported that the oncogenic MYC transcription factor directly regulates expressions of a number of splicing regulating proteins, leading to multiple oncogenic splicing changes[17–19]. However, the relationships between the pre-mRNA splicing/spliceosome and other oncogenes/tumor suppressors are largely unknown.

Tumor suppressor PTEN (phosphatase and tensin homolog on chromosome 10) acts as a bona fide dual lipid and protein phosphatase[20,21]. The most extensively studied tumor suppressive function of PTEN is its lipid phosphatase activity, by which it dephosphorylates the PtdIns(3,4,5)P3 (PIP3) to PIP2, thereby depleting cellular PIP3, a potent activator of AKT[20–22]. However, cells harboring phosphatase-inactive PTEN mutants retain residual tumor suppressive activity[23–25]. Now, it is believed that cytoplasmic PTEN is primarily involved in regulating phosphatidylinositol-3-kinase (PI3K)/PIP3 signaling, while nuclear PTEN exhibits phosphatase-independent tumor suppressive functions, including regulation of chromosome stability, DNA repair and apoptosis[25–29]. Thus, the systematical identification of phosphatase-independent functions of PTEN may provide new insights into the strategies targeting PTEN-deficient cancers[30–33]. However, the mechanisms through which non-catalytic activities of PTEN contribute to its tumor suppressor function are still poorly understood.

Here, we show that nuclear PTEN can interact with the spliceosomal proteins and drive pre-mRNA splicing in a phosphatase-independent manner. In particular, PTEN depletion promotes Golgi extension and secretion through GOLGA2 exon skipping. These results suggest that Golgi secretion inhibitors alone or in combination with PI3K/Akt kinase inhibitors may be therapeutically useful for PTEN-deficient cancers.

## Results

**PTEN regulates global AS**. To investigate whether PTEN plays a role in splicing, we transiently transfected the adenovirus E1A reporter plasmid pMTE1A as a minigene (Fig. 1a)[34] or the-double-reporter plasmid pTN24 constitutively expressing β-galactosidase (β-gal), in which luciferase is expressed only if appropriate splicing removes an upstream intron that contains translational stop codons (Fig. 1b)[35] into PTEN[+/+] and PTEN[−/−]mouse embryonic fibroblast (MEF) cells. The results demonstrated that PTEN deficiency promoted selection of the most proximal 5' splice site with increase of 13S isoform and decrease of 12S/9S isoforms of pMTE1A minigene (Fig. 1a) or caused a decreased splicing ratio of pTN24 (Fig. 1b), as detected by the decreased luciferase/β-gal ratio. These results propose that PTEN regulates minigene splicing.

To evaluate the potential impacts of PTEN on AS, total RNAs from 293T cells with or without PTEN depletion by two pairs of small hairpin RNAs (shRNAs; shPTEN#1 or shPTEN#2 vs nonspecific shRNA as negative control (shNC)) were detected by RNA sequencing (RNA-seq). More intriguingly, 262 AS events (ASEs) representing 208 human genes were detected to be significantly changed under PTEN knockdown by two pairs of shRNAs (Fig. 1c, d and Supplementary Data 1), among which the most frequently occurring were cassette exons (CA) with much more events of exon exclusion (EX) than inclusion (IN), and mutually excluded exons (MXE), alternative 3' and 5' acceptor sites (A3SS and A5SS), were also included (Fig. 1e). Some of the ASEs were validated by reverse-transcription polymerase chain reaction (RT-PCR; Fig. 1f). All these experiments strongly support the correlation of PTEN and ASEs.

**PTEN-regulated ASEs are implicated in cancers**. To identify whether PTEN-regulated ASEs occur in human tumors, we applied the pipeline used on 293T cells to tumor collections from The Cancer Genome Atlas (TCGA) (http://cancergenome.nih.gov). Given the common role of PTEN as a tumor suppressor and the variation of ASEs among different cancer types, all available cancer types were initially considered. ASEs with >5% of frequency in a cancer type were considered to be cancer related, and the results showed that 80 out of 262 PTEN-regulated ASEs identified in 293T cells were cancer related (Fig. 2a and Supplementary Data 2). Of note, these ASEs were most significantly enriched with high frequencies in two collections of cancer samples, cervical squamous cell carcinoma and endocervical adenocarcinoma (CESC) and glioblastoma multiforme and low-grade glioma (GBMLGG) (Fig. 2a), the latter having been reported to harbor the highest deletion/mutation frequencies in PTEN gene among all cancer types[36]. Thus, we sought to elaborate the relationship between PTEN-regulated ASEs and cancer by using the GBMLGG collection. Among the 262 PTEN-regulated ASEs identified in 293T cells, we identified 20 cancer-related ASEs in GBMLGG. We divided these cancer samples into two groups with or without PTEN copy loss, and identified 20 ASEs to be correlated with the status of PTEN copy number (Fig. 2b and Supplementary Table 1). Also, 21 ASEs were identified to be significantly correlated with overall survival of patients, either favorable or unfavorable (Supplementary Fig. S1A and Supplementary Table 2). Based on these analyses, we found that 60% (12/20) of PTEN-correlated ASEs were also cancer related (Fig. 2c). To our astonishment, 16 out of the 20 PTEN-correlated ASEs also correlated with patient survival, all of which predicted significant worse outcome (Fig. 2d, e). These analyses strongly support that PTEN-regulated ASEs have functional implications in cancers.

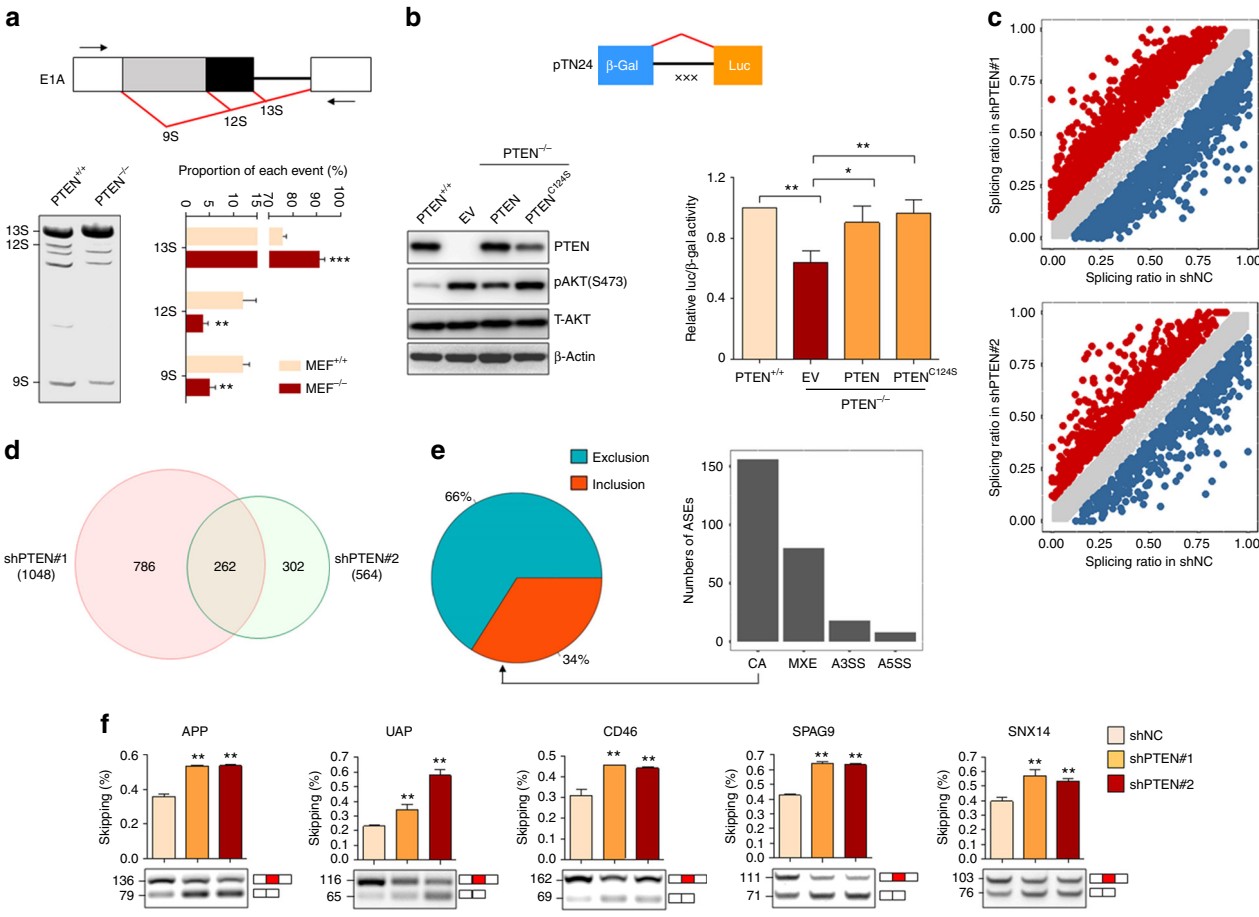

**Fig. 1** PTEN regulates alternative splicing. **a** Quantification of E1A mRNA isoforms. A diagram of the E1A reporter gene indicating the alternative 5′ splice sites and splicing events that generate 13S, 12S and 9S mRNAs. The locations of the exon primers used for RT-PCR analysis are shown (top). RT-PCR analysis of pMTE1A containing E1A reporter gene transfected in PTEN$^{+/+}$ and PTEN$^{-/-}$ MEF cells was performed, with a representative example (left, bottom) and quantification for the percentage of each isoform from three independent biological replicates (right, bottom) are shown. **b** The pTN24 minigene construct (top) consisting of β-gal, an upstream intron that contains three translational stop codons (represented as "×××") and luciferase (Luc), was transfected into PTEN$^{+/+}$ MEF cells, as well as PTEN$^{-/-}$ MEF cells infected with empty vector (EV), PTEN$^{WT}$ or PTEN$^{C124S}$. The indicated proteins were immunoblotted (left, bottom), and the ratios of luciferase expression relative to β-gal expression are shown (right, bottom). **c** Splicing ratio profiles of the ASEs identified in 293T cells with or without PTEN knockdown. The red and blue dots respectively represent significantly upregulated and downregulated ASEs in shPTEN#1 (top) or shPTEN#2 (bottom)-infected cells compared to shNC ones. **d** The Venn diagram of ASEs significantly changed in shPTEN#1 and shPTEN#2 compared to shNC-infected cells. **e** Numbers of four kinds of ASEs (right) and numbers of excluded (EX) or included (IN) in CA (left) significantly changed in PTEN knocked down 293T cells. **f** RT-PCR analysis of five representative PTEN-regulated EX splicings identified in 293T cells. Quantification of three independent biological replicates (top) and a representative example is shown (bottom). For **a**, **b**, **g**, data represent means with bar as s.d. of three independent experiments; *$p < 0.05$; **$p < 0.01$; ***$p < 0.001$; two-sided unpaired $t$-test

There are 10 ASEs presented in all three subgroups (Fig. 2f) which were considered to be the PTEN-regulated ASEs most relevant to cancers, and all of which show negative correlation with patient survival (Fig. 2g and Supplementary Fig. 1B). To choose candidates for in-depth functional analysis, the occurrence of each ASE was validated by using the glioblastoma cell lines SF188 with intact PTEN and U251 with deficient PTEN. In total, 3 ASEs, corresponding to three genes, *GOLGA2*, *KIF21A* and *C2CD5*, were faithfully validated in U251 versus SF188 cells, shPTEN#1 versus shNC-infected SF188 cells, and PTEN or its phosphatase-defective mutant (PTEN$^{C124S}$, which had no inhibitory effect on AKT activation[21]) versus empty vector (EV)-expressing U251 cells (Supplementary Fig. 1C).

**Nuclear PTEN interacts with spliceosomal proteins**. The above-described findings promoted us to investigate how PTEN regulates ASEs. As depicted in Fig. 1b, re-expression of either wild-type PTEN (PTEN$^{WT}$) or PTEN$^{C124S}$ in PTEN$^{-/-}$ MEF cells reversed the inhibitory effect of PTEN depletion on pTN24 splicing. In agreement, PTEN phosphatase inhibitors failed to recapitulate or reverse the effect of PTEN depletion on pTN24 splicing (Supplementary Fig. 2A, B). All of these results proposed that PTEN regulates AS in a phosphatase-independent manner. To consolidate this, we stably transduced PTEN$^{WT}$- or PTEN$^{C124S}$-expressing plasmids, both of which are synonymous mutants resistant to shPTEN#1, into 293T-shPTEN#1 cells. As expected, the ectopic expression of PTEN$^{WT}$ and PTEN$^{C124S}$ effectively reversed the shPTEN#1-induced ASEs to the similar degree (Supplementary Data 3 and Supplementary Fig. 2C, D). These results further support that the phosphatase-independent activities of PTEN contribute to its regulation on the AS.

Next, we subjected nuclear extracts of 293T cells to size exclusion chromatography and the resultant chromatographic fractions were analyzed for PTEN. As demonstrated in Fig. 3a, the majority of nuclear PTEN protein was detected in the fractions

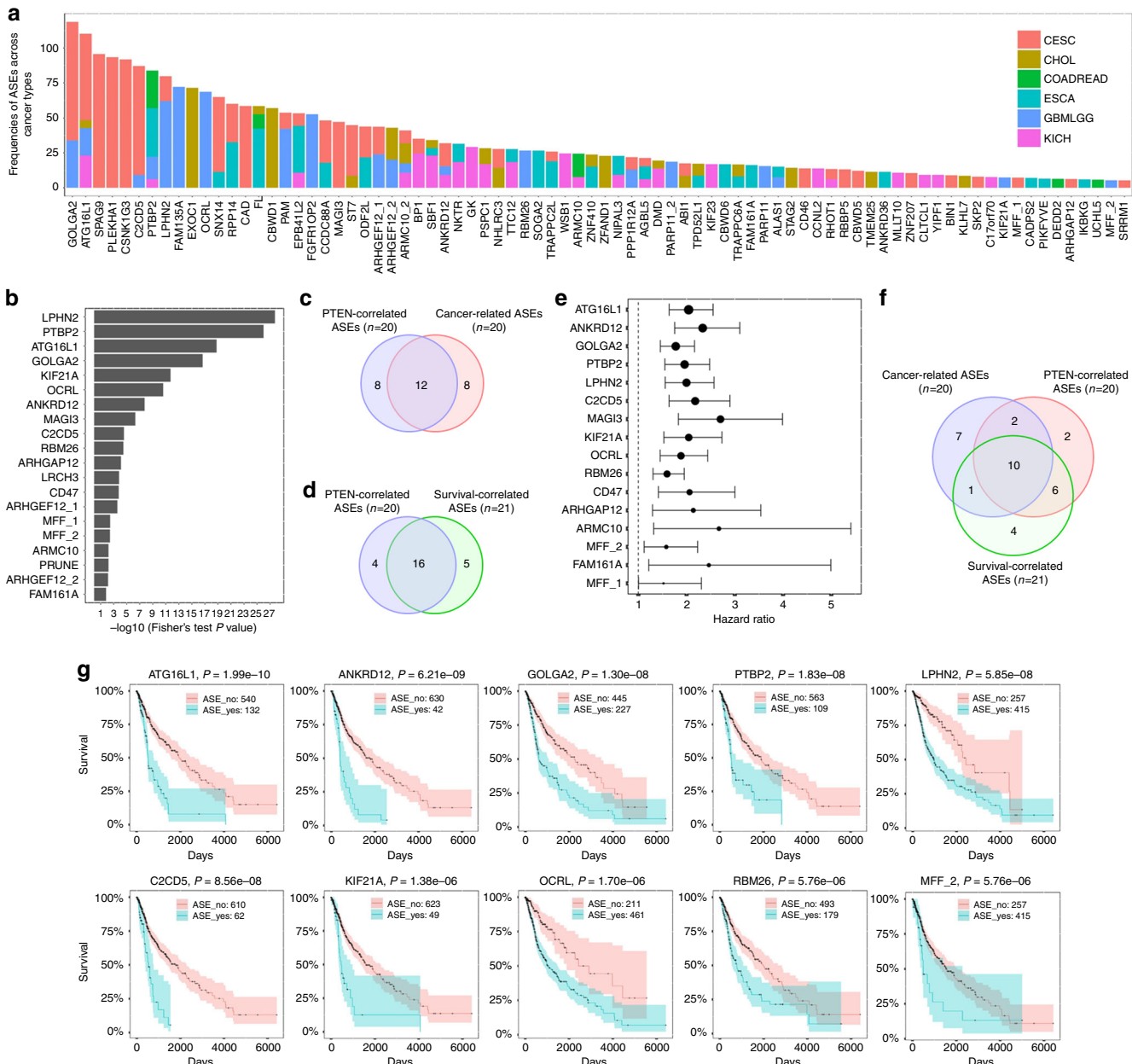

**Fig. 2** PTEN-regulated ASEs are cancer-related and correlate with patient survival. **a** The 262 PTEN-regulated ASEs identified in 293T cells were analyzed in TCGA tumor collections. ASEs with frequency of >5% in a single cancer type were considered to be cancer related, and the frequencies of cancer-related ASEs across 6 cancer types are shown. CESC cervical squamous cell carcinoma and endocervical adenocarcinoma, CHOL cholangiocarcinoma, COADREAD colon adenocarcinoma+rectum adenocarcinoma, ESCA esophageal carcinoma, GBMLGG glioblastoma multiforme+low-grade glioma, KICH kidney chromophobe. **b** Tumor samples of GBMLGG were divided into two groups with or without PTEN copy loss, and 20 ASEs were identified to be correlated with PTEN status according to the frequency of each ASE in the two groups as assayed by Fisher's exact test. **c** The Venn diagram of cancer-related ASEs from **a** and PTEN-correlated ASEs from **b** in GBMLGG. **d** The Venn diagram of PTEN-correlated ASEs from **b** and survival-correlated ASEs from Supplementary Fig. 1A in GBMLGG. **e** Hazard ratios of PTEN- and survival-correlated ASEs from **d**. **f** The Venn diagram of cancer-related ASEs from **a**, PTEN-correlated ASEs from **b**, and survival-correlated ASEs from Supplementary Fig. 1A in GBMLGG. **g** Tumor samples of GBMLGG were divided into two groups with or without occurrence of an ASE, and survival curves were drawn between the two groups for each ASE from **f**

around 50 kDa, corresponding to the monomeric form of PTEN protein. Intriguingly, a portion of PTEN protein also appeared in fractions with greater molecular mass (>699 kDa), proposing that nuclear PTEN might be recruited to macromolecular complexes. To address this, nuclear extracts of 293T cells were subjected to affinity purification using an anti-PTEN antibody with non-specific IgG as negative control, and the bound proteins were analyzed by liquid chromatography–tandem mass spectrometry (LC–MS/MS). Two independent experiments were performed,

and 136 unique proteins were identified in both experiments (Supplementary Fig. 3A and Supplementary Data 4). The KEGG (Kyoto Encyclopedia of Genes and Genomes) Pathway analysis revealed that these PTEN-interacting proteins were involved in spliceosome, mRNA surveillance, herpes simplex infection, RNA polymerase, RNA transport and ribosome (Fig. 3b). Especially, 38 out of 136 proteins are listed as components in the spliceosome (Fig. 3b), including components from three major snRNPs (U1, U2 and U5), the EJC/TREX, the PRPF19–CDC5L complexes,

and two classes of RNA-binding proteins, serine/arginine-rich (SR) proteins and heterogeneous nuclear ribonucleoproteins (hnRNPs) (Fig. 3c).

To confirm the association between PTEN and spliceosomal proteins, cytoplasmic and nuclear proteins were extracted from 293T cells (Fig. 3d), MEF (Fig. 3e) or prostate cancer cell line DU145 cells (Supplementary Fig. 3B), and co-

immunoprecipitations (co-IPs) with anti-PTEN antibody or non-specific IgG were performed, followed by immunoblotting with antibodies against some spliceosomal proteins. As expected, the abundance of nuclear PTEN was relatively lower compared to that of cytoplasmic PTEN, and most of the spliceosomal proteins detected were mainly localized in nuclei, although some of them also appeared in the cytoplasm (Fig. 3d, e and Supplementary

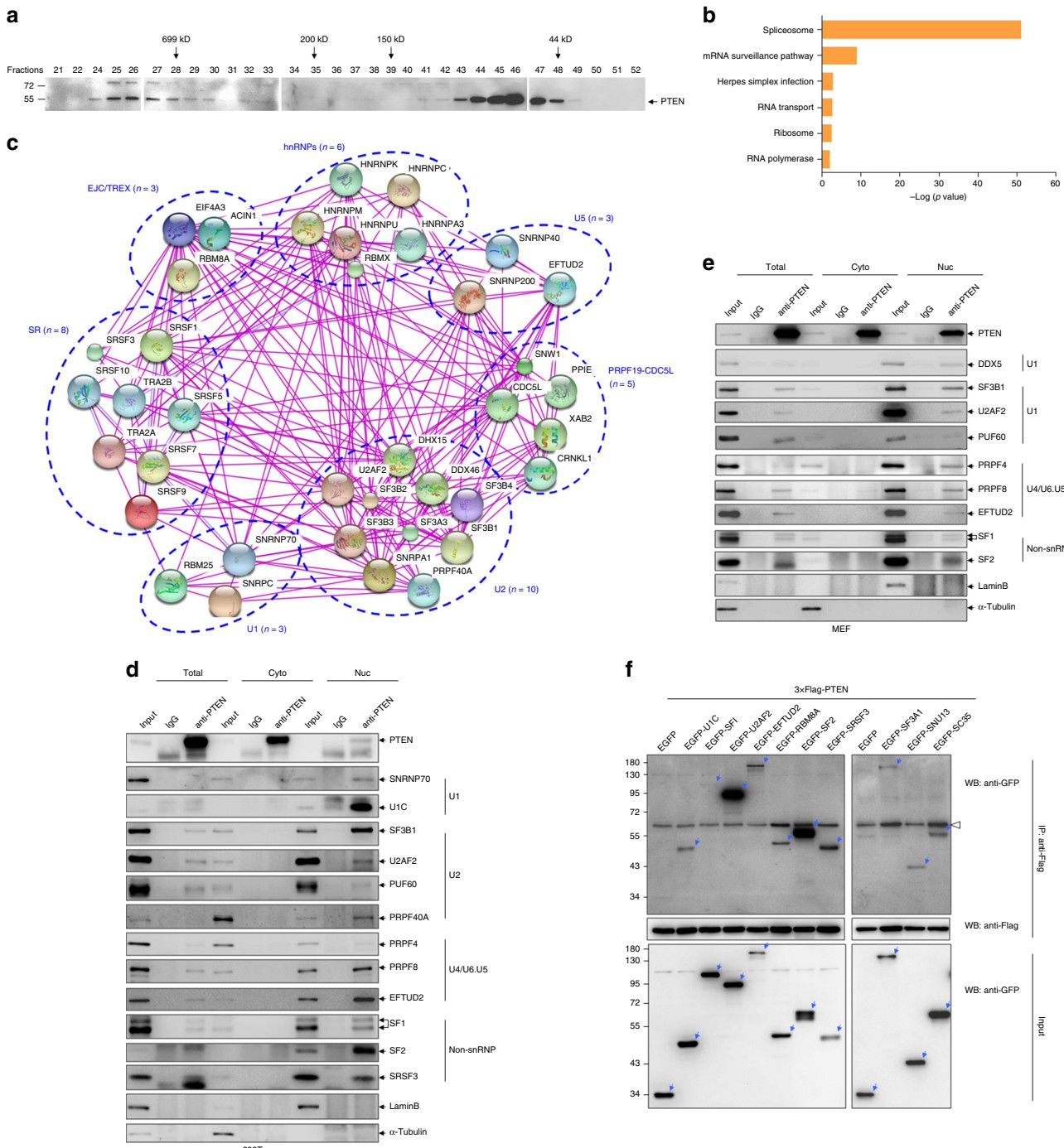

**Fig. 3** PTEN associates with spliceosomal proteins. **a** Size exclusion chromatography analysis of nuclear extracts from 293T cells, followed by immunoblotting with anti-PTEN antibody. **b** KEGG pathway analyses of PTEN-associated proteins. **c** Analysis of the PTEN-associated spliceosomal proteins by String database. The edge indicates known interaction between two proteins. Proteins in the same complex or family are circled. **d**, **e** 293T (**d**) and MEF (**e**) cells were fractionated into total, cytoplasmic (Cyto) and nuclear (Nuc) fractions, and immunoprecipitation with anti-PTEN antibody or control IgG was performed in each fraction, followed by immunoblotting for indicated proteins. **f** 3×Flag-tagged PTEN was cotransfected with EGFP-tagged spliceosomal proteins or EGFP into 293T cells, and immunoprecipitation with anti-Flag antibody was performed, followed by immunoblotting with anti-Flag or anti-GFP antibody. The blue arrows points to the corresponding EGFP-tagged proteins, and empty arrow indicates a non-specific band

Fig. 3B). More importantly, anti-PTEN antibody precipitated the spliceosomal proteins detected to different extents in whole cell lysates and nuclear fractions of all three kinds of cells (Fig. 3d, e and Supplementary Fig. 3B), confirming an association between PTEN and spliceosome in the nuclei. We also showed that ectopically expressed 3×Flag-tagged PTEN pulled down all 10 kinds of green fluorescent protein (GFP)-tagged spliceosomal proteins in 293T cells (Fig. 3f), supporting the interaction of PTEN with spliceosomal proteins.

**PTEN directly interacts with RS domain of U2AF2.** Because PTEN deficiency did not significantly change the expressions of splicing factors tested (Supplementary Fig. 4A), we speculated that PTEN might play a role during spliceosome assembly. Since spliceosome assembly occurs co-transcriptionally within the chromatin environment, the abundances of chromatin-associated snRNAs have been used to study the dynamic assembly of

snRNPs into the spliceosome[37]. Thus, quantitative RT-PCR (qRT-PCR) was applied to detect the chromatin association of all five spliceosomal snRNAs in MEF cells. Interestingly, only chromatin abundance of U2 snRNA was constantly and substantially decreased upon PTEN depletion in MEF cells (Fig. 4a), suggesting that PTEN loss causes reduced recruitment of U2 snRNP during spliceosome assembly, also mirroring the above finding that components from U2 snRNP overwhelmingly associate with PTEN.

To find the direct mechanism underlying PTEN-regulated U2 snRNP recruitment during spliceosome assembly, we tried to identify direct interactors of PTEN in spliceosome, especially in U2 snRNP. To rule out the possibility that the association of PTEN with spliceosome is mediated through RNA, nuclear extracts from 293T cells were treated with RNase, which removes RNA and thus abolishes RNA-mediated interactions, and co-IP assays showed that all the spliceosomal proteins tested could be pulled down with anti-PTEN antibody to the similar extent in

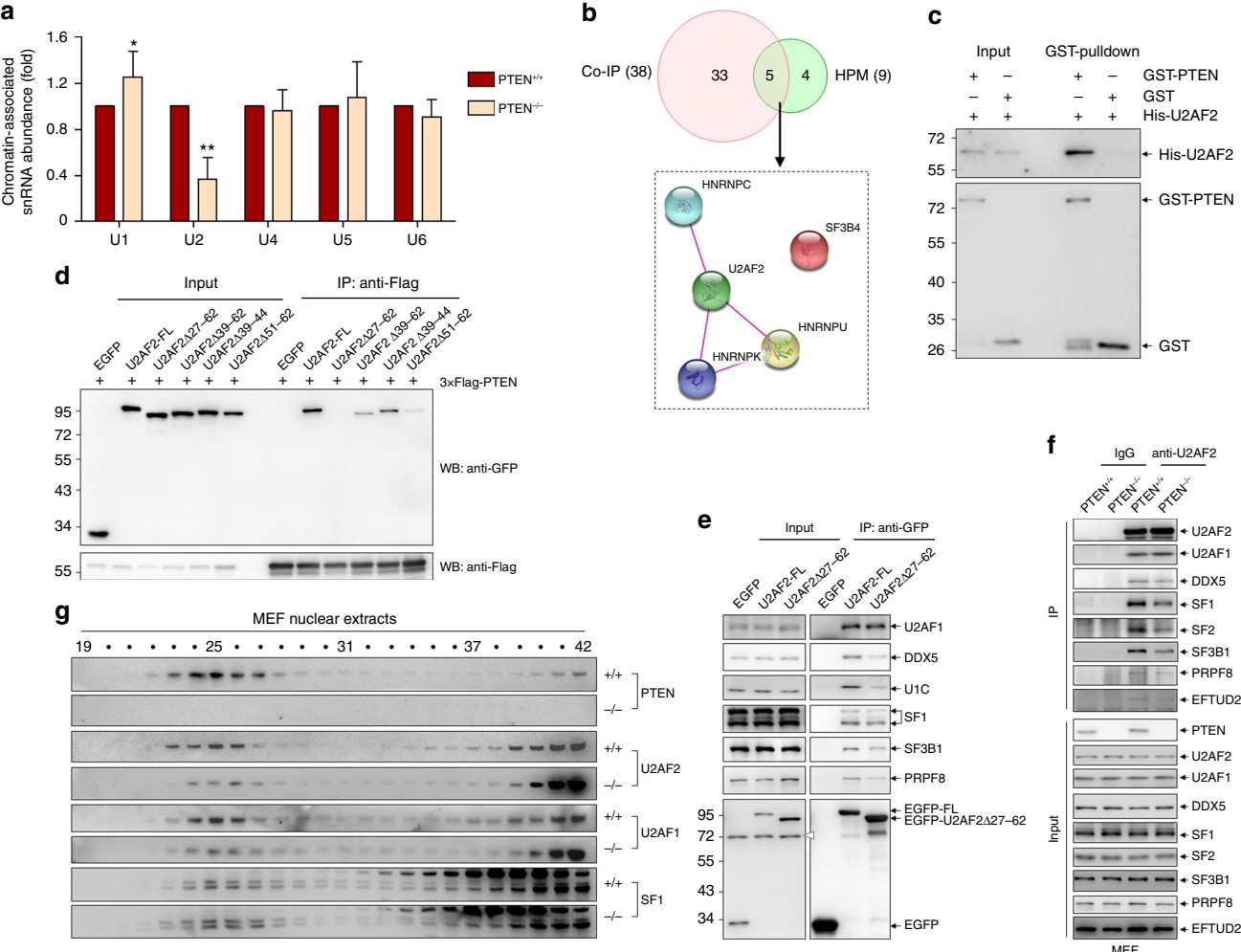

**Fig. 4** PTEN regulates spliceosome assembly through U2AF2. **a** Chromatin-associated five snRNAs in PTEN[+/+] and PTEN[−/−] MEF cells were assayed by qRT-PCR. Data represent means with bar as s.d. of three independent experiments; *$p < 0.05$; **$p < 0.01$; two-sided unpaired t-test. **b** The Venn diagram of PTEN-interacting spliceosomal proteins identified by co-IP assay or HPM (top). Spliceosomal proteins identified by both assays are analyzed by String database (bottom). The edge indicates known interaction between two proteins. **c** Bacterially expressed GST or GST-tagged PTEN proteins were incubated with His-tagged U2AF2, followed by GST pulldown and immunoblotting for His and GST. **d** 3×Flag-tagged PTEN and EGFP-tagged U2AF2 constructs were cotransfected into 293T-PTENΔ cells, and immunoprecipitation with anti-Flag antibody was performed, followed by immunoblotting with anti-Flag or anti-GFP antibody. **e** EGFP, EGFP-tagged U2AF2-FL or U2AF2Δ27-62 was transfected into 293T-PTENΔ cells, and immunoprecipitation with GFP-Trap was performed, followed by immunoblotting for indicated proteins. **f** Immunoprecipitation with anti-U2AF2 antibody or control IgG in PTEN[+/+] and PTEN[−/−] MEF cells, followed by immunoblotting for indicated proteins. **g** Size exclusion chromatography analysis of nuclear extracts from PTEN[+/+] and PTEN[−/−] MEF cells, followed by immunoblotting with fractions 19–42 from both cells for indicated proteins

both the presence and absence of RNase (Supplementary Fig. 4B), suggesting that PTEN−spliceosome association mainly involved protein−protein interaction. Thus, we probed a human proteome microarray (HPM) consisting of 20,240 individual N-terminally glutathione S-transferase (GST)-tagged proteins with biotinylated PTEN protein. Briefly, bacterially expressed PTEN was purified and biotinylated, and the resultant PTEN−biotin conjugates were incubated with the HPM, and proteins with PTEN-binding capacity were identified by adding Cy5-conjugated streptavidin (Supplementary Fig. 4C). Using the stringent criteria as described in the Methods section, 712 proteins were identified as potential PTEN-interacting proteins, including the known PTEN-interacting proteins USP7 (HAUSP7)[38], WWP2[39] and APC3 (CDC27)[25] (Supplementary Data 5 and Supplementary Fig. 4D). More interestingly, the KEGG Pathway analysis indicated that 9 spliceosomal proteins also appeared as potential PTEN-interacting proteins (Supplementary Fig. 4E), among which 5 proteins, U2AF2, SF3B4, HNRNPC, HNRNPK and HNRNPU (Fig. 4b), were also found in the aforementioned co-IP assay (Fig. 3c), suggesting that PTEN can directly interact with at least these five spliceosome proteins. Among those five PTEN-binding proteins, U2AF2 is generally accepted to play a vital role in the recruitment of U2 snRNP to the spliceosome[40]. Hence, we decided to confirm the potential direct interaction of U2AF2 with PTEN with GST pulldown assay. Indeed, bacterially expressed GST-tagged PTEN but not GST alone pulled His-tagged U2AF2 down (Fig. 4c), indicating these two proteins can interact directly with each other.

There are several well-characterized motifs in U2AF2 that serve to bind proteins or RNAs, as indicated in Supplementary Fig. 5A. To define the domain of U2AF2 for its interaction with PTEN, we expressed these enhanced green fluorescent protein (EGFP)-tagged N-terminal (U2AF2-N), middle (U2AF2-M) and C-terminal (U2AF2-C) U2AF2 segments (Supplementary Fig. 5A) besides its full length (U2AF2-FL) for co-IP assay with 3×Flag-tagged PTEN in 293T cells depleted of endogenous PTEN by CRISPR−CAS9. The results showed that N terminus of U2AF2 is required for its interaction with PTEN (Supplementary Fig. 5B). We continued to refine the PTEN-interacting motif of U2AF2, and found that aa27–62 bound PTEN with similar strength to the U2AF2-FL, while the other parts of U2AF2 showed no binding (Supplementary Fig. 5C, D). Reciprocally, the deletion of aa27–62 completely abolished U2AF2−PTEN interaction (Supplementary Fig. 5E). Furthermore, progressive deletion of aa27–62 showed that only deletion of the whole domain (U2AF2Δ27-62) completely abolished U2AF2−PTEN interaction, while deletion of aa39–62 (U2AF2Δ39-62) or aa51–62 (U2AF2Δ51-62) greatly diminished it (Fig. 4d). Therefore, the aa27–62, corresponding to a part of the arginine-serine-rich (RS) domain of U2AF2, mediates U2AF2–PTEN interaction. It was reported that phosphorylation of multiple serines in the RS domains of other proteins may play important roles in the AS[41]. Thus, we also decided whether phosphorylation in the RS domain of U2AF2 affects its interaction with PTEN. To this end, we analyzed the RS domain of U2AF2 with Scansite tool which suggests that seven serines (top, Supplementary Fig. 5F) are potential phosphorylation sites, and co-IP assay showed that PTEN failed to bind the phosphomimetic RS domain mutant of U2AF2 by substituting these seven serines with aspartic acids (Supplementary Fig. 5F), suggesting that phosphorylation of U2AF2 in the RS domain might disrupt the interaction between PTEN and U2AF2. On the other hand, the fragment compassing aa150–286 of PTEN, which mapped to its C-terminal part of the phosphatase domain and N-terminal part of the C2 domain, is required for its interaction with U2AF2 (Supplementary Fig. 6).

**PTEN depletion inhibits U2AF2 recruitment into spliceosome.** To date, the function of the RS domain has not been studied so extensively as the other motifs of U2AF2, especially in the cellular context. We transfected EGFP-tagged U2AF2-FL and U2AF2Δ27-62 into 293T cells, and found that the U2AF2Δ27-62 mutant was not as efficiently recruited to the spliceosome as U2AF2-FL, as evidenced by the significantly attenuated interactions between the U2AF2 mutant and other spliceosomal proteins, except for U2AF1 (Fig. 4e), indicating that the PTEN-binding RS domain plays a role in the recruitment of U2AF2 to the spliceosome.

We continued to investigate whether PTEN affects the recruitment of U2AF2 to spliceosome. As for this, endogenous U2AF2-associated proteins were immunoprecipitated from PTEN$^{+/+}$ and PTEN$^{-/-}$ MEF cells using an anti-U2AF2 antibody. Although U2AF2 levels were comparable and there was no obvious difference in the U2AF2−U2AF1 interaction between PTEN$^{+/+}$ and PTEN$^{-/-}$ cells, the PTEN$^{-/-}$ cells displayed an attenuated association between U2AF2 and other spliceosomal proteins tested (Fig. 4f). Therefore, PTEN is required for the efficient recruitment of U2AF2 into spliceosome. In view of this, we speculated that the co-localization of U2AF2 with other spliceosomal proteins should be lost upon PTEN depletion. Indeed, size exclusion chromatography with nuclear extracts showed that U2AF2, U2AF1 and SF1 proteins appeared together with endogenous PTEN in the >699 kDa fractions in PTEN$^{+/+}$ MEF cells, while the amounts of U2AF2 and U2AF1 but not SF1 in the macromolecular fractions were dramatically reduced, accompanied with a concomitant increase in the low molecular mass fractions in PTEN$^{-/-}$ MEF cells (Fig. 4g). Taken together, PTEN loss disrupts the recruitment of U2AF2, which probably in turn undermines the assembly of U2 snRNP into spliceosome.

**PTEN depletion induces GOLGA2 alternative splicing.** We selected GOLGA2 to exemplify the role of PTEN-regulated ASEs in carcinogenesis. GOLGA2 encodes the Golgi matrix protein GM130, a peripheral membrane component of the *cis*-Golgi stack that acts as a membrane skeleton to maintain the Golgi apparatus (GA) structure, and a vesicle tether to facilitate vesicle fusion to the Golgi membrane[42]. The canonical isoforms of human (NM_004486.4) and mouse (NM_001080968.1) GOLGA2 consists of 26 exons. One isoform that lacks exons 1–14 in humans and one isoform that lacks exon 3 in mice for GOLGA2 have been previously found. Our RNA-seq analyses revealed that 293T cells expressed a previously undescribed GOLGA2 isoform with the inclusion of an extra exon (here designated as exon 2b) between exon 2 and exon 3 (Fig. 5a), which is named GOLGA2$^L$ isoform, and PTEN loss promoted the exclusion of exon 2b. We thus used RT-PCR to confirm the results of RNA-seq with primers spanning exon 2b (Fig. 5a). As shown in Fig. 5b, GOLGA2$^L$ was more abundantly expressed than the exon 2b-skipped isoform designated as GOLGA2$^S$ in all seven cell lines detected. PTEN loss resulted in the increase of GOLGA2$^S$ to a significant degree in all these cells. Notably, PTEN did not appear to be the sole decisive factor of GOLGA2 exon 2b splicing, because PTEN depletion did not result in complete loss of GOLGA2$^L$ and PTEN-intact cells expressed considerable amounts of GOLGA2$^S$. Also, the imbalance in the AS induced by PTEN loss is variable in different cell lines. Moreover, U2AF2 knockdown also induced GOLGA2 exon 2b skipping (Supplementary Fig. 7A), supporting the co-regulation of exon 2b by PTEN and U2AF2.

**PTEN loss promotes GA extension and secretion via GOLGA2 skipping.** To find out the consequence of GOLGA2

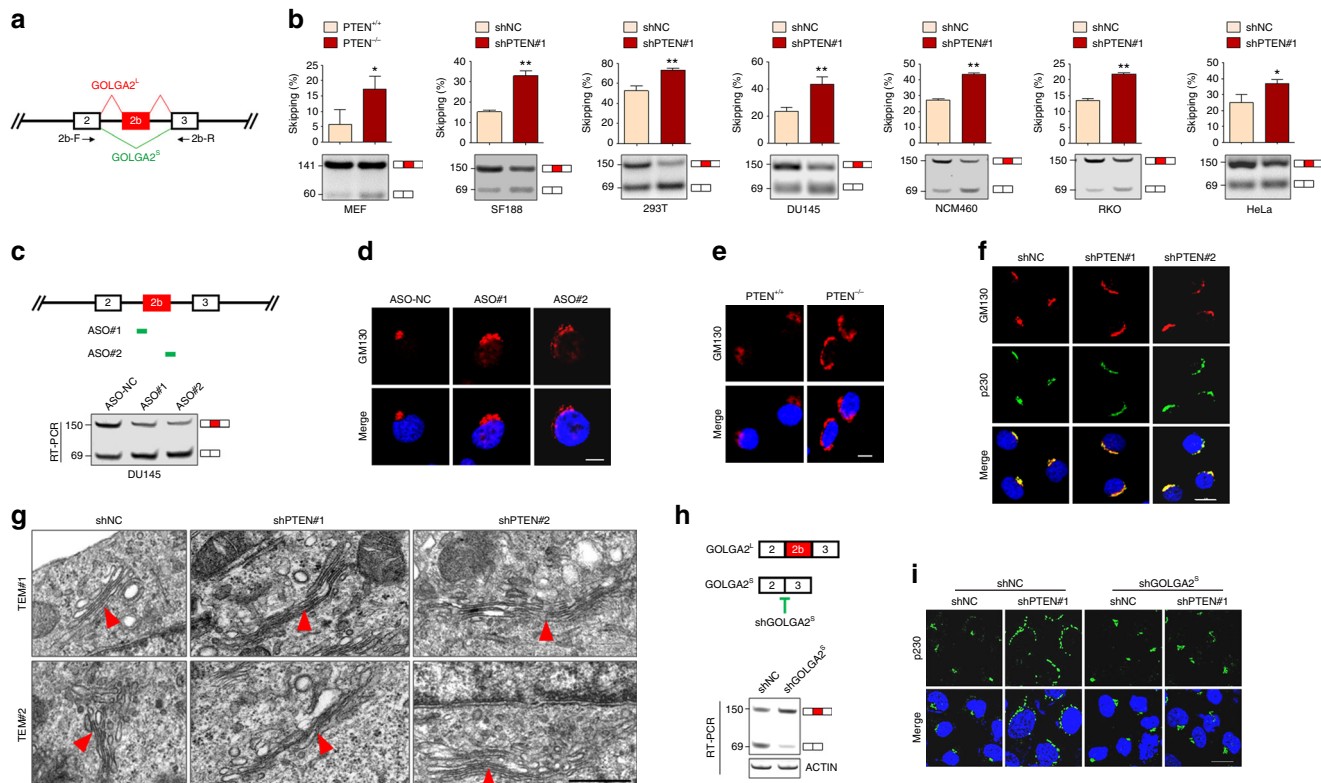

**Fig. 5** PTEN regulates Golgi extension through GOLGA2 exon 2b splicing. **a** Schematic representation of a part of GOLGA2 gene with exons 2, 2b and 3 shown as boxes. Splicing patterns are shown as diagonal red (exon 2b included) or green (exon 2b excluded) lines. Arrows indicate the locations of primers used in RT-PCR analysis. **b** RT-PCR analysis of GOLGA2 exon 2b alternative splicing in PTEN$^{+/+}$ and PTEN$^{-/-}$ MEF cells, as well as in the indicated cells with or without PTEN knockdown. Quantification of three independent biological replicates (top) and a representative example (bottom) are shown. Data represent means with bar as s.d. of three independent experiments; *$p < 0.05$; **$p < 0.01$; two-sided unpaired $t$-test. **c** Schematic representation of ASOs designed to induce exon 2b skipping of GOLGA2 (top), and RT-PCR verification of GOLGA2 exon 2b skipping in ASOs-transfected DU145 cells. **d**, **e** Immunofluorescent staining of GM130 together with re-staining of DAPI in ASOs-transfected DU145 cells (**d**) and PTEN$^{+/+}$ and PTEN$^{-/-}$ MEF cells (**e**). Scale bar represents 10 μm. **f** Immunofluorescent staining of GM130 and p230 together with re-staining of DAPI in DU145 cells with or without PTEN knockdown. Scale bar represents 20 μm. **g** Two fields of TEM images of Golgi structure of DU145 cells with or without PTEN knockdown. The red triangles point to Golgi. Scale bar represents 500 nm. **h** A pair of shRNA targeting the exon 2/3 junction for specific knockdown of GOLGA2$^S$ was designed (top) and transfected into DU145 cells, and the efficiency was verified by RT-PCR (bottom). **i** DU145 cells with or without GOLGA2$^S$ knockdown were further subjected to PTEN knockdown, and immunofluorescent staining of p230 together with re-staining of DAPI were performed. Scale bar represents 20 μm

exon 2b splicing, we electroporated two steric hindrance antisense oligonucleotides (ASOs) into DU145 and NCM460 cells. These ASOs resulted in the specific induction of GOLGA2 exon 2b skipping in both DU145 (Fig. 5c) and NCM460 (Supplementary Fig. 7B) cells. Immunofluorescent staining with anti-GM130 antibody showed that ASO-expressing DU145 (Fig. 5d) and NCM460 (Supplementary Fig. 7C) cells exhibited more extended distribution of perinuclear Golgi staining. This was also true in PTEN$^{-/-}$ MEF cells (Fig. 5e) and in PTEN knocked down DU145 (Fig. 5f), SF188 and RKO cells (Supplementary Fig. 7D). The same results could also be seen in PTEN$^{-/-}$ MEF cells by staining of Golgi-localized Golgin-97 (Supplementary Fig. 7E). The extended distribution of GA in PTEN$^{-/-}$ MEF cells could also be confirmed by transient transfection with EGFP-tagged PH domain from FAPP1 (Supplementary Fig. 7F), which exhibits specific localization to Golgi membranes[43]. Furthermore, transmission electron microscopy (TEM) observations clearly demonstrated that PTEN$^{-/-}$ MEF cells presented the longer Golgi cisternae with reduced cisternae thickness compared with PTEN$^{+/+}$ MEF cells (Supplementary Fig. 7G), a morphological change that resembled Golgi extension[44]. The similar

phenomenon could also be seen in DU145 cells upon PTEN knockdown (Fig. 5g). Moreover, specific knockdown of the GOLGA2$^S$ by shGOLGA2$^S$, which efficiently reduced the level of GOLGA2$^S$ by targeting the exon 2/3 junction (Fig. 5h), almost completely reversed the phenotypic effect of PTEN depletion on Golgi extension (Fig. 5i), supporting that the ability of PTEN depletion to promote Golgi extension is dependent on the expression of GOLGA2$^S$.

The mRNA sequence of GOLGA2$^S$ matches to that of canonical GOLGA2, whose protein product is GM130[45]. Theoretically, exon 2b (81 bp) inclusion will not result in frameshift or premature termination, meaning that GOLGA2$^L$ should encode a protein product with extra 27 amino acids in the N terminus of GM130 and with ~3 kD difference in molecular weight compared to GM130 (Supplementary Fig. 8A). Unexpectedly, the antibody recognizing the C terminus of GM130, which should detect protein products of both GOLGA2$^S$ (GM130) and GOLGA2$^L$, only detected a single band in several cell lines subjected to western blot (Supplementary Fig. 8B). Interestingly, the level of this band was increased by PTEN knockdown in all three cell lines tested (Supplementary Fig. 8B). To decide the

identity of this band, we transfected DU145 and 293T cells with shRNAs respectively targeting GOLGA2$^S$ or GOLGA2$^L$. As shown in Supplementary Fig. 8C, both shRNAs efficiently knocked down their intended targets without affecting the other. However, only shRNA targeting GOLGA2$^S$ significantly reduced

the band detected by anti-GM130 antibody, while knockdown of GOLGA2$^L$ did not affect the band. We then asked whether GOLGA2$^L$ encoded a protein product with very low stability. For this purpose, cells were treated with MG132, but no accumulated band was observed (Supplementary Fig. 8D). Thus, we proposed

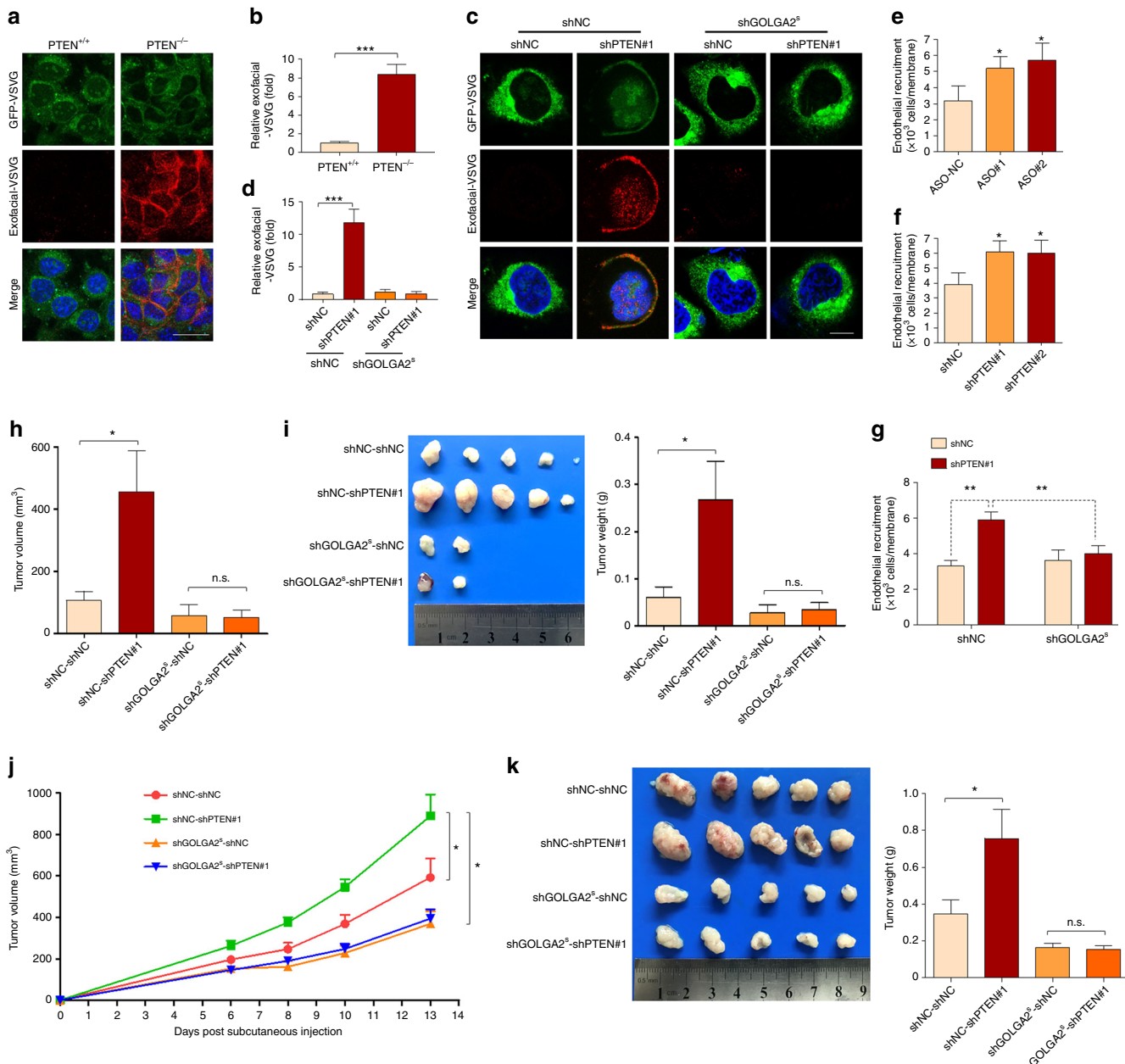

**Fig. 6** GOLGA2$^S$ promotes secretion and tumorigenesis upon PTEN loss. **a, b** Trafficking of ts045-VSVG-EGFP in PTEN$^{+/+}$ and PTEN$^{-/-}$ MEF cells. Cells transfected with ts045-VSVG-EGFP were incubated at 40 °C for 3 h, then shifted to 32 °C for 1 h. Arrival at plasma membrane of ts045-VSVG-EGFP was determined with antibody against the exofacial-VSVG in unpermeabilized cells with scale bar representing 25 μm (**a**). The ratio of plasma membrane-arrived VSVG was determined by comparing exofacial-VSVG signal to the total cellular EGFP signal, followed by normalization to the control group (**b**). **c, d** DU145 cells with or without GOLGA2$^S$ knockdown were subjected to PTEN knockdown, and trafficking of ts045-VSVG-EGFP were determined (**c**) with scale bar representing 10 μm and quantified (**c**). **e–g** DU145 cells transfected with ASOs (**e**), with or without PTEN knockdown (**f**) and with or without GOLGA2$^S$ knockdown together with shNC or shPTEN#1 viruses (**g**) were subjected to endothelial recruitment assay. **h, i** DU145 cells with or without GOLGA2$^S$ knockdown were infected with shNC or shPTEN#1 viruses, and subcutaneously injected into nude mice. After 52 days, tumor volumes were measured (**h**), harvested and weighed (**i**). **j, k** SW620 cells with or without GOLGA2$^S$ knockdown were infected with shNC or shPTEN#1 viruses, and subcutaneously injected into nude mice. Tumor volumes were measured at different time points (**j**). At 13 days after subcutaneous injection, tumors were harvested and weighed (**k**). For **b** and **d**, data represent means with bar as s.d. of 10 cells in each group. For **e–g**, data represent means with bar as s.d. of three independent experiments. For **h–k**, data represent means with bar as s.d.; *p < 0.05; **p < 0.01; ***p < 0.001; two-sided unpaired t-test for **b**, **d**, **e–i**, **k**; two-sided paired t-test for **j**

that canonical GM130 encoded by GOLGA2$^S$ constituted the majority of the protein products of GOLGA2 isoforms, and PTEN knockdown increased GM130 by promoting GOLGA2 exon 2b skipping.

**PTEN deficiency promotes tumorigenesis via GOLGA2 skipping.** To find out whether PTEN depletion affects the Golgi secretory capacity, we examined trafficking in cells depleted of PTEN by using the anterograde cargo ts045-VSVG-EGFP, which has been widely used to study membrane transport because of its reversible misfolding and retention in the endoplasmic reticulum (ER) at 40 °C and its ability to move out of the

ER into the Golgi complex, and eventually to the plasma membrane upon temperature reduction to 32 °C. To this end, MEF cells transfected with ts045-VSVG-EGFP were incubated at 40 °C, then shifted to 32 °C. Arrival at the plasma membrane of ts045-VSVG-EGFP was determined with antibody against the extracellular domain of vesicular stomatitis virus G (VSVG; named exofacial-VSVG) in unpermeabilized cells, allowing unambiguous detection of VSVG on the plasma membrane[46]. The results showed that PTEN$^{+/+}$ MEF cells had little exofacial-VSVG on plasma membrane, which became more significant in PTEN$^{-/-}$ MEF cells (Fig. 6a, b), suggesting that PTEN depletion robustly increased trafficking of ts045-VSVG-EGFP to the plasma

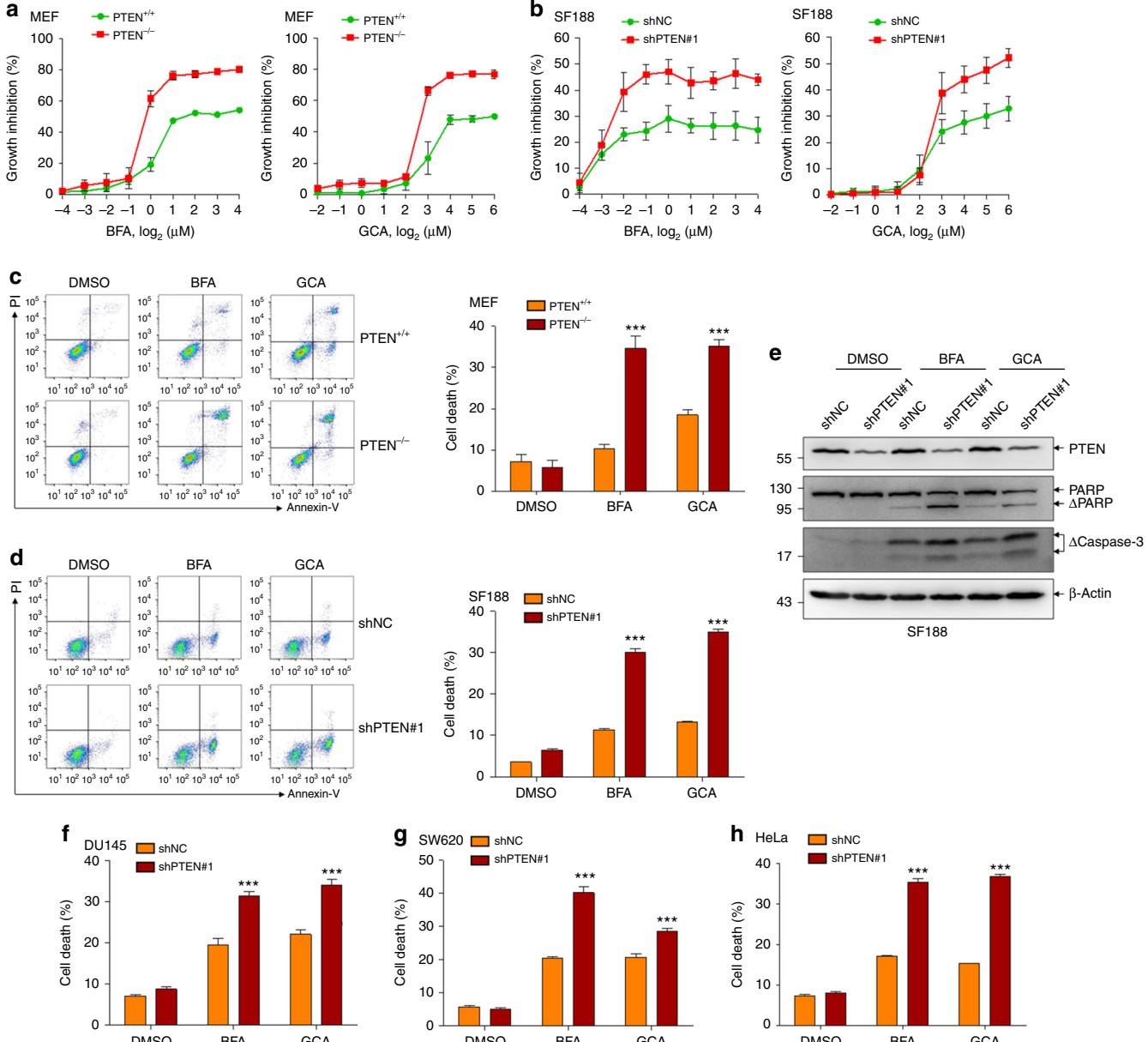

**Fig. 7** PTEN deficiency sensitizes cancer cells to secretion inhibitors. **a** Growth inhibition of PTEN$^{+/+}$ and PTEN$^{-/-}$ MEF cells at 24 h after treatment with the indicated concentrations of BFA or GCA. **b** Growth inhibition of SF188 cells with or without PTEN knockdown at 14 h after treatment with the indicated concentrations of BFA or GCA. **c** Representative FACS plots of annexin-V and propidium iodide (PI) staining (left) and quantification of cell death (right) of PTEN$^{+/+}$ and PTEN$^{-/-}$ MEF cells following 24 h of BFA (1 μM), GCA (10 μM) or DMSO treatment. **d, e** SF188 cells with or without PTEN knockdown were treated for 14 h with BFA (1 μM), GCA (10 μM) or DMSO treatment. Representative FACS plots of annexin-V and PI staining (left, **d**), quantification of cell death (right, **d**) and immunoblotted for indicated proteins (**e**) are shown. **f–h** DU145 (**f**), SW620 (**g**) and HeLa (**h**) cells with or without PTEN knockdown were stained with annexin-V and PI following 32, 28 and 46 h of BFA (1, 1.5, 2 μM), GCA (10, 10, 15 μM) or DMSO treatment. Quantification of cell death is shown. Data represent means with bar as s.d. of three independent experiments; ***$p$ < 0.001; two-sided unpaired $t$-test

membrane in MEF cells. In agreement, knockdown of PTEN also increased the trafficking of VSVG in DU145 cells, which was inhibited by the knockdown of GOLGA2$^S$ (Fig. 6c, d). Secretion-dependent endothelial recruitment assays[44] also revealed that electroporation of ASOs (Fig. 6e) and PTEN knockdown (Fig. 6f) significantly enhanced endothelial recruitment abilities of DU145 cells, which was greatly inhibited by knockdown of GOLGA2$^S$ (Fig. 6g).

To investigate the contribution of GOLGA2 splicing in PTEN depletion-enhanced tumorigenesis, DU145 cells with or without GOLGA2$^S$ depletion were further subjected to PTEN knockdown, and subcutaneous tumor growth was monitored. As shown in Fig. 6h, i, the advantage of PTEN depletion on tumor growth was observed on tumors without knockdown of GOLGA2$^S$, but was diminished by knockdown of GOLGA2$^S$. Of note, knockdown of basal level of GOLGA2$^S$ inhibited tumor growth. Similar results were obtained in SW620 cells (Fig. 6j, k), whose subcutaneous growth is much faster than DU145 as well as other cell lines used in Fig. 5b. To investigate whether PTEN can reverse GOLGA2$^S$-induced tumor growth, we selected U251 cells which expressed high GOLGA2$^S$ isoform due to PTEN deficiency (Supplementary Fig. 1C) to restore the expression of PTEN$^{WT}$ or PTEN$^{C124S}$. As expected, the re-expressions of both forms of PTEN could significantly inhibit the tumor growth in U251 cells (Supplementary Fig. 9A, B). Taken together, GOLGA2 splicing contributes to PTEN loss-of-function-induced secretion and tumor growth.

**PTEN depletion sensitizes cancer cells to secretion inhibitors**. Finally, PTEN$^{+/+}$ and PTEN$^{-/-}$ MEF cells were treated by two commonly used secretion inhibitors brefeldin A (BFA) and golgicide A (GCA). The results showed that, once a certain concentration was reached (1 μg/ml for BFA and 8 μg/ml for GCA), both BFA and GCA inhibited PTEN$^{-/-}$ cells to a greater extent than PTEN$^{+/+}$ cells (Fig. 7a). Similar phenomenon was also observed on BFA- and GCA-treated SF188 cells (Fig. 7b), indicating PTEN depletion sensitizes cells to growth inhibition induced by secretion inhibitors. Further, both BFA and GCA treatments resulted in significantly more cell death in PTEN-depleted MEF and SF188 (Fig. 7c–e) cells. Similar results could also be recapitulated in other cancer cell lines from different origins, including DU145 (Fig. 7f), SW620 (Fig. 7g) and HeLa (Fig. 7h) cells.

## Discussion

Applying nuclear extracts of 293T cells to size exclusion chromatography, here we found that nuclear PTEN proteins appeared in fractions with much greater molecular mass than its monomeric form, although PTEN dimerization can occur in both the nucleus and cytoplasm[47]. Therefore, it is more important to identify nuclear PTEN interactome for deepening its roles. We performed co-IP-based proteomic analysis in nuclear extracts of 293T cells, and identified 38 components in the spliceosome as PTEN-interacting proteins. We also used bacterially expressed biotinylated PTEN to probe a HGM, and found 712 potential PTEN-interacting proteins, which was more significant than the previous report[48]. Of great importance, 9 spliceosomal proteins including U2AF2 were shown to be potential PTEN-interacting proteins. Here we confirmed that PTEN and U2AF2 proteins indeed directly interact with each other, for which the RS domain of U2AF2 is absolutely required, although how other several components interact with PTEN remains to be investigated. More intriguingly, the RS domain is also required for the efficient recruitment of U2AF2 to spliceosome, although the domain does not overlap with the motifs required for the binding of SF1/SF3b1 (UHM), U2AF1 (ULM) or Py-tract.

However, PTEN had no impetus on expressions of spliceosomal proteins. On the other hand, AS sites are generally weaker, and suboptimal binding may render them particularly sensitive to levels of U2AF. Actually, U2AF2 recruitment appears to be a highly regulated process[40,49]. The binding of U2AF2 to Py-tract is strengthened by cooperative protein−protein interactions with SF1 and U2AF1[40]. U2AF heterodimer can be subjected to competition or enhancement by posttranslational lysyl-5-hydroxylation[50] and other RNA-binding proteins[51–53]. More intriguingly, we found that PTEN deletion inhibits the recruitment of U2AF2 and subsequent U2 snRNP to spliceosome during spliceosome assembly. In parallel, the RS domain-deleted U2AF2 mutant was not efficiently integrated into the spliceosome. These findings propose that PTEN enhances spliceosome assembly through its interaction with the RS domain of U2AF2, although its exact mechanisms remains to be further investigated.

U2AF heterodimer has the capacity to directly define ~88% of functional 3' splice sites (ss) in the human genome, but numerous U2AF binding events also occur in intronic locations[49,54]. Further mechanistic dissection reveals that upstream intronic binding events interfere with the immediate downstream 3' ss associated either with the alternative exon, to cause exon skipping, or with the competing constitutive exon, to induce exon inclusion[49]. Of note, the reduced recruitment of U2AF2 does not necessarily mean diminished splicing of pre-mRNA. A previous report showed that addition of excessive SC35 protein can functionally substitute for U2AF2 in the reconstitution of pre-mRNA splicing in U2AF2-depleted extracts[55]. Therefore, it is possible that there exist a variety of complementary mechanisms or factors to substitute for U2AF2. Thus, we speculated that PTEN loss does not simply attenuates splicing, but might also lead to a dysregulated splicing program. Indeed, PTEN deletion or knockdown significantly interfered AS of two minigene splicing reporters. Our further investigations showed that PTEN loss globally changed AS of many genes in human cells. Such effects of PTEN are phosphatase independent, because they could be restored by re-introduction of the phosphatase-defective PTEN mutant. The roles of these altered splicing in the initiation and progression of nuclear PTEN deletion or mutation-related cancer and their potentials as a therapeutic target specifically against PTEN-deficient cancer would be an area of continued investigations.

To our knowledge, PTEN has never been connected with GA. As a central organelle in the secretory pathway of cells, the GA connects upstream (ER) or downstream (endosome, lysosome and PM) compartments and plays important roles in the posttranslational modification, sorting and transportation of all newly synthesized secretory proteins as well as transmembrane proteins from the ER. This secretory pathway is being intensively studied in the development of novel targets for anticancer therapies[56]. As one of the consequences resulting from PTEN deficiency-induced aberrant splicing, AS of GOLGA2 provided us the opportunity to take a preliminary look at the relationship between PTEN and GA. We showed that PTEN deletion caused GOLGA2 exon 2b skipping, resulting in increased GM130 protein, although the mechanisms remain to be investigated. Whether GOLGA2$^L$ transcript give rise to any protein product remained to be further confirmed by developing an antibody against the 27 amino acid insert of GOLGA2$^L$. However, we could not rule out the possibility that GOLGA2$^L$ might act as a non-coding RNA. Actually, several protein-coding genes have also been reported to generate long non-coding RNA isoforms by AS to play a distinct role[57]. Whether GOLGA2$^L$ also acts as a non-coding RNA will definitely be interesting for us to explore in our future studies. As a peripheral membrane protein strongly attached to the Golgi membrane, GM130 facilitates vesicle fusion to the Golgi membrane as a vesicle "tethering factor" together with p115, giantin and

GRASP65. GM130 is also involved in the maintenance of the Golgi structure and plays a major role in the disassembly and reassembly of the Golgi apparatus during mitosis[58], and is involved in the control of glycosylation, cell cycle progression and higher order cell functions such as cell polarization and directed cell migration, as previously reviewed[45]. Indeed, PTEN loss morphologically causes Golgi extension. Functionally, the secretory capacity of GA is dramatically enhanced by PTEN loss. Such morphological and functional changes of GA have been proved to promote metastasis capacity of cancer cells, consistent with the metastasis-suppressive role of PTEN. Of note, the PTEN-splicing–GA axis is only one example that the role of PTEN can be extended to unexpected realms through altering the splicing program. Given the extensive range of genes that are impacted by splicing, in our opinion, the function of PTEN is still largely to be explored.

## Methods

**Cell culture.** $Pten^{-/-}$ MEFs were a gift from Professor Hongbing Zhang[59]. Human umbilical vein endothelial cells (HUVECs) were kindly provided by Professor Ping-Jin Gao at the Institutes of Health Sciences (Shanghai, China)[60]. The 293T, U251, SW620, RKO, HeLa and DU145 cells were purchased from Cell Bank of Chinese Academy of Sciences, Shanghai. NCM460 cells were purchased from Shanghai HONSUN Biological Technology Co., Ltd. SF188 cells were obtained from the University of California, San Francisco[61]. SW620 and NCM460 cells were maintained in RPMI-1640 supplemented with 10% fetal bovine serum (FBS) and cultured in a humidified incubator at 37 °C with 5% CO₂. All other cell lines were maintained in Dulbecco's modified Eagle's medium (DMEM) supplemented with 10% FBS. There were no signs of mycoplasma contamination for all cell lines. Cell line authentication was performed via short tandem repeat profiling.

**Antibodies.** Antibodies employed are shown in Supplementary Table 3.

**Plasmids and shRNAs.** Plasmid expressing 3×Flag-tagged PTEN was generated by inserting PTEN CDS into pQCXIN vector. Plasmids expressing EGFP-tagged proteins were generated by inserting correspondent CDS into pEGFP-C1 vector. CDS of spliceosomal proteins was kindly provided by Professor Jiahuai Han's lab (Xiamen University, Xiamen, China). Plasmid expressing GST-tagged PTEN was generated by inserting PTEN CDS into pEGX-4T-3 vector. Plasmids expressing His-tagged proteins were generated by inserting correspondent CDS into pET-28a (+) vector. The pMTE1A plasmid was a gift from Adrian R. Krainer (Cold Spring Harbor Laboratory, Cold Spring Harbor, New York). The pTN24 plasmid was a gift from Ian C. Eperon (University of Leicester, Leicester, UK). Plasmid carrying GOLGA2 CDS with or without exon 2b was generated by inserting correspondent CDS into pBABE vector. pEGFP-VSVG was obtained from Addgene (Cambridge, MA, plasmid #11912). The pLVX-IRES-ZsGREEN1-PTEN and -PTEN$^{C124S}$ plasmids have been previously described[21]. For rescuing experiments in shPTEN#1-infected cells, synonymous mutations of 4 nucleotides in the CDS of both PTEN and PTEN$^{C124S}$ was performed with the primers listed in Supplementary Table 4. Plasmids for knockdown were constructed by inserting corresponding shRNA sequences into the pSIREN Retro-Q plasmid (Clontech). The shRNA sequences specially targeting PTEN are GATCTTGACCAATGGCTAAGT (shPTEN#1) and CGGGAAGACAAGTTCATGTACTT (shPTEN#2). The shRNA sequence targeting GOLGA2$^S$ is ACCTGAGGATACACCCAAGGA (shGOLGA2$^S$). The shRNA sequence targeting GOLGA2$^L$ is GGATATTCAGGACATTCTGAA (shGOLGA2$^L$). The shRNA sequence targeting U2AF2 is CGCCTTCTGTGAGTACGTGGA. All constructs were verified by DNA sequencing.

**Subcellular fractionation.** Cells were lysed in hypotonic buffer (10 mM Tris-HCl, pH 7.9, 1.5 mM MgCl₂, 10 mM KCl, 1 mM dithiothreitol (DTT), 1 mM phenylmethylsulfonyl fluoride (PMSF) and 1× protease inhibitor cocktail (Calbiochem)) by a Dounce homogenizer (40 strokes). Nuclear pellets were separated from cytoplasm by centrifugation for 10 min at 1000 × g. The supernatants (cytoplasmic extract) were removed and transferred into new tubes. Nuclear pellets were washed with hypotonic buffer twice, and resuspended in the nuclear extraction buffer (50 mM Tris-HCl, pH 7.9, 1 mM MgCl₂, 1 mM DTT, 0.1% Nonidet P-40 (NP-40), 250 units/ml Benzonase (Sigma), 1 mM PMSF and 1× protease inhibitor cocktail (Calbiochem)) by sonication. The supernatants (nuclear extracts) were collected by centrifugation for 10 min at 17,000 × g.

**Immunoprecipitation.** Cells were harvested and lysed with immunoprecipitation buffer (50 mM Tris-HCl, PH 7.6, 150 mM NaCl, 1 mM EDTA, 1% NP-40, 1 mM PMSF, and 1× protease inhibitor cocktail (Calbiochem)). After brief sonication, the supernatants (whole cell lysates) were collected by centrifugation at 12,000 × g for 10 min at 4 °C. For immunoprecipitation of Flag or GFP-tagged proteins, whole cell lysates were incubated with anti-Flag M2 Affinity Gel (Sigma-Aldrich) or GFP-Trap (ChromoTek) at 4 °C overnight. Otherwise, supernatants (cytoplasmic extracts, nuclear extracts or whole cell lysates) were incubated with indicated antibodies overnight and protein A/G-agarose beads (Santa Cruz, CA) for 4 h at 4 °C. The precipitates were washed three times with immunoprecipitation buffer, boiled in sample buffer and subjected to immunoblot assay. RNase A treatment was at 100 μg/ml for 30 min at room temperature. Primary antibodies used were shown in Supplementary Table 3.

**Nano-LC–ESI-MS/MS analysis.** Immunoprecipitation samples were separated by sodium dodecyl sulfate–polyacrylamide gel electrophoresis, and visualized with colloidal Coomassie blue. The lane from gels was cut into 1 mm slices, and each slice was washed twice with 50 mM NH₄HCO₃, 50% acetonitrile (ACN) and dehydrated with ACN. Proteins were reduced and alkylated with 10 mM DTT and 55 mM iodoacetamide, respectively. After washing with 50 mM NH₄HCO₃ and ACN, proteins were digested in gel with trypsin (Promega, Madison, WI) and incubated overnight at 37 °C. Tryptic peptides were extracted from the gel pieces with 60% ACN and 0.1% trifluoroacetic acid. The peptide extracts were vacuum centrifuged to dryness. Dried peptides were dissolved to 10 μl of 2% ACN and 0.1% trifluoroacetic acid. Samples were desalted and preconcentrated through a Michrom peptide CapTrap (MW 0.5-50 kDa, 0.5 × 2 mm; Michrom BioResources, Inc., Auburn, CA). The eluent was vacuum centrifuged to dryness then reconstituted in 2% ACN and 0.1% formic acid.

For LC–MS/MS analysis, 10 μl samples were introduced from an autosampler (HTS-PAL, CTC Analytics, Zwingen, Switzerland) at a flow of 1 μl/min for 15 min onto a reverse-phase microcapillary column (0.1 × 150 mm, packed with 5 μm 100 Å Magic C18 resin; Michrom Bioresources) using a HPLC (Paragram MS4, Michrom Bioresources). The reverse-phase separation of peptides was performed at a flow of 0.5 μl/min using the following buffers: 2% ACN with 0.1% formic acid (buffer A) and 98% ACN with 0.1% formic acid (buffer B) using a 90 min gradient (0-35% B for 90 min, 80% B for 8 min, 95% B for 12 min and 0% B for 20 min). The eluate was introduced directly onto a hybrid linear ion trap (LTQ) Orbitrap mass spectrometer (ThermoFinnigan, San Jose, CA, USA) equipped with ADVANCE Spray Source (Michrom Bioresources). A high-resolution MS survey scan was obtained for the $m/z$ 350–1800, $R = 100,000$ (at $m/z$ 400) and ion accumulation to a target value of $10^6$. Siloxane ($m/z$ 445.120025) was used as an internal standard. MS/MS spectra were acquired using data-dependent scan from the 10 most intense ions with charge states ≥2 in the survey scan (as determined by the Xcalibur mass spectrometer software in real time). Only MS signals exceeding 500 ion counts triggered a MS/MS attempt and 5000 ions were acquired for a MS/MS scan. Dynamic mass exclusion windows of 27 s were used. Singly charged ions and ions with unassigned charge states were excluded from triggering MS/MS scans. The normalized collision energy was set to 35%. All MS/MS ion spectra were extracted by ProteoWizard msConvert and were analyzed using Mascot (Matrix Science, London, UK; version 2.4.1). Mascot was set up to search the uniprot_human database (88,625 entries) assuming the digestion enzyme semiTrypsin allowed for two missed tryptic cleavages with full mass from 600 to 4600. Mascot was searched with a fragment ion mass tolerance of 1.00 Da and a parent ion tolerance of 10.0 PPM. Carbamidomethyl of cysteine was specified in Mascot as a fixed modification. Oxidation of methionine and acetyl of the N terminus were specified in Mascot as variable modifications. Scaffold (version Scaffold_4.2.1, Proteome Software Inc., Portland, OR) was used to validate MS/MS-based peptide and protein identifications by the Scaffold Local FDR algorithm. Protein identifications were accepted if they could be established at greater than 95.0% probability. Protein probabilities were assigned by the Protein Prophet algorithm. Proteins that contained similar peptides and could not be differentiated based on MS/MS analysis alone were grouped to satisfy the principles of parsimony. Proteins sharing significant peptide evidence were grouped into clusters.

**Size exclusion chromatography.** Size fractionation analysis was carried out using a Superdex 200 column (GE Healthcare). Protein standards of 699, 440, 200, 150, 44 and 29 kDa were used to calibrate the column in 10 mM Tris-HCl, pH 7.4, 500 mM NaCl, 1 mM EDTA. One milligram (~1.0 ml) of nuclear extract was loaded onto the Superdex 200 column, and 0.3 ml fractions were collected. Immunoblot analyses using 20 μl from each fraction were performed.

**Human Proteome Microarray.** The recombinant His-PTEN fusion proteins were labeled with Biotin (Full Moon Biosystems) and used to probe the ProtoArray Human Protein Microarray (Wayen Biotechnologies). Briefly, ProtoArray slides were pre-blocked with blocking buffer, and then incubated for 1 h at room temperature with 5 μg/5 ml Biotin-labeled His-PTEN, with gentle shaking. Slides were then washed three times with 1× phosphate-buffered saline with Tween-20 (PBST) and two times with distilled water. The microarrays were further incubated with Cy5-conjugated streptavidin (1‰) for 20 min at room temperature and underwent five more 5 min washes. The microarrays were spun dry and subjected to scanning with a GenePix 4000B (Axon Instruments, Sunnyvale, CA) in order for results to be visualized and recorded. A GenePix Pro 6.0 was used for data analysis. The data were extracted by GenePix 6.0 from the microarray images. To generate the candidate list of PTEN-binding proteins, the signal-to-noise ratios (SNR = (F635

mean − B635 mean)/B635 s.d.) and another index, the fold change (F635 median/B635 median), were firstly calculated for all the spots. The cutoff was set as SNR ≥3 and fold change ≥5. To determine the final candidates, proteins with coefficient of variation of <0.15 from two duplicated spots were considered.

**Luciferase/β-gal double-reporter assay.** The 293T cells were seeded in a 12-well plate ($8 \times 10^4$ cells per well). After culturing overnight, cells were transfected with the pTN24 splicing reporter plasmid (containing a constitutively expressed β-galactosidase reporter for transfection normalization and a luciferase reporter that is conditional on removal of a translational stop codon by splicing). In some experiments, cells were also cotransfected with PTEN-specific siRNA or siRNA scramble control. Cells were harvested 48 h after transfection, and reporter expression was detected as previously described[62] using the Dual Light Reporter System (Applied Biosystems) and analyzed by calculating the ratio of the luciferase to β-galactosidase signals. Three independent experiments were performed, and the data were analyzed using Student's *t*-test.

**Recombinant expression.** Recombinant full-length PTEN and U2AF2 were expressed in *Escherichia coli* using the PET-28a vector expression system and purified by nickel chelating, ion exchange and gel filtration columns as reported. Recombinant proteins were kept in 10 mM Tris-HCl, pH 7.8,150 mM NaCl, 2 mM DTT.

**GST pulldown assays.** We expressed and purified GST and GST-tagged PTEN proteins as previously described[63]. The 15 μg His-tagged U2AF2 was mixed with 4 μg GST or GST-PTEN bound to glutathione-sepharose beads in 0.5 ml of binding buffer. The binding reaction was performed overnight at 4 °C, and the beads were subsequently washed three times with the binding buffer.

**Western blots.** Cell lysates were separated by SDS-polyacrylamide gel, transferred to nitrocellulose membrane (Bio-Rad, Richmond, CA), blocked by 5% nonfat milk in tris-buffered saline and immunoblotted with the indicated antibodies. Followed by incubation with horseradish peroxidase (HRP)-linked second antibody (Cell Signaling, Beverly, MA) at room temperature for 1 h, detection was performed by Immobilon Western Chemiluminescent HRP substrate kit (Merck Millipore) according to the manufacturer's instructions. Primary antibodies used are shown in Supplementary Table 3. Uncropped images of important immunoblots are presented in Supplementary Figure 10.

**Immunofluorescence.** Coverslip-grown cells were fixed in 4% paraformaldehyde for 15 min at room temperature, and rinsed three times in PBS. Coverslips were permeabilized in 0.3% Triton X-100/PBS for 15 min and rinsed three times in PBS. Then, coverslips were blocked in 2% bovine serum albumin/PBS for 1 h at room temperature. Primary antibodies were applied in a 1:100 dilution in staining buffer overnight at 4 °C in a humid chamber. Coverslips were subsequently washed three times. Secondary antibodies (Alexa Fluor secondary 488, 595; Invitrogen) were applied in a 1:200 dilution in staining buffer for 1 h at 37 °C in a humid chamber in the dark. Prior to mounting with Vectorshield with 4',6-diamidino-2-phenylindole (DAPI; Vector Laboratories, CA), coverslips were washed three times more in PBS. Immunofluorescence analysis was conducted on a Nikon Eclipse TI Laser Scanning Microscope or Leica TCS SP8. Image analysis was performed using ImageJ software. Primary antibodies used are shown in Supplementary Table 3.

**RNA sequencing and ASE analysis.** Paired-end sequencing (150 base pairs) was performed on Illumina Hi-Seq X-Ten platform. Reads were mapped to the National Center for Biotechnology Information (NCBI) GRCh37 human genome or mm9 mouse genome assemblies using STAR v2.5. Gene and isoform expression quantification was called by RSEM v1.2 with default parameters on the gene annotation file. Differential alternative splicing events were discovered based on the STAR splicing junction file and isoform expression levels (see below). A pipeline is developed to detect differential alternative splicing events between RNA-seq samples of treatment and control conditions. It is based on a metric defined as splicing ratio (SR) which was used in some splicing-related studies[64,65]. The SR for a junction in a sample is calculated here as the ratio of the read count involving that junction to the sum of read counts of junctions sharing the same upstream 5' splice site (5' SR) or the same downstream 3' splice site (3' SR). An alternative splicing event is composed of two splicing junctions sharing the same 5' splice site or 3' splice site involving three exons: 5' end upstream exon, inclusion exon and skipping exon. Types of alternative splicing events were identified from relative genomic positions of the three exons. Briefly, the pipeline has two major steps. First, both 5' SR and 3' SR for all splicing junctions of samples are calculated. For each treatment sample, a set of candidate differential splicing events are chosen by comparing with all the control samples. A score variable srd (splicing ratio difference) defined in the equation $\frac{1 + \sum_{i=1}^{N_c}(SR_{tj} - SR_{ci})}{1 + \frac{s.d.(SR_c)}{N_c}}$, where for a splicing event, $SR_c$ is splicing ratios across control samples and $SR_{tj}$ is the splicing ratio of *j*th treatment samples, s.d. is for standard deviation and $N_c$ is the number of control samples. It is used to estimate the extent to which a splicing event is changed in

experiment condition versus control condition. The distribution of srd values in a treatment sample is fitted by the heavy-tailed distribution. A big srd value in the tail indicates that a splicing event is likely significantly changed in the treatment condition. The significance levels of srd values for splicing events in a sample can be estimated by using R package poweRlaw. The median of significance values across treatment samples is used to select candidate differential splicing events. The threshold can be dynamically set according to the background distribution of significance values calculated from control samples. Second, in order to limit false positives while maximize true positives, candidate differential splicing events are sorted by expression levels of isoforms containing inclusion exons or skipping exons. It is because big srd values may be caused by low gene/isoform expression levels and sequencing depth/bias. Splicing events are sorted by the sum of absolute differences of inclusion isoforms and skipping isoforms between treatment and control conditions. This pipeline was demonstrated to have a higher precision than existing tools for knock-out data sets by randomly selecting top splicing events and experimentally validating them.

**CRISPR/Cas9.** HEK293T-*PTEN*Δ cells were generated using CRISPR/Cas9. Briefly, guide RNA (gRNA) sequence targeting the particular locus in the second exon of PTEN (target sequence: 5'-ACATTATTGCTATGGGATTTC-3') were ordered as complementary primers, mixed in a 1:1 ratio and annealed. Subsequently, pAAV-U6-gRNA-mCMV-SaCAS9-P2A-sfGFP (Obio Technology, Shanghai, CHINA) was digested using *Bsm*BI and the gRNA was introduced using a normal ligation reaction according to the manufacturer's instructions (New England Biolabs). Cells were infected with virus containing the gRNA and 48 h later GFP-positive cells were selected prior to plating for individual clones. Homozygous editing of the PTEN loci was confirmed by PCR and sequencing. Furthermore, expression of a PTEN mRNA with frameshift deletion in the second exon was confirmed by RT-PCR and sequencing the complementary DNA (cDNA).

**qRT-PCR and RT-PCR.** Total RNA was prepared using Trizol (Invitrogen) following the manufacturer's instructions. Following digestion with DNase I (Promega), 2 μg RNA was subjected to reverse transcription to synthesize cDNA using random primers (Takara) and M-MLV Reverse Transcriptase (Promega, Fitchburg, WI). qRT-PCR was performed using the SYBR Green PCR Master Mix (Applied Biosystems, Foster City, CA). Primers are shown in Supplementary Table 4. Novel ASOs were applied to nascent transcripts of a target gene via Watson-Crick bonding to exert steric hindrance effect against splicing factors to modulate splicing. The ASOs were rationally designed for optimal efficiency in inducing splicing modulation. Briefly, ASO target sites were selected by a computational algorithm (ESE finder, human splicing finder) that accounted for co-transcriptional binding accessibilities, binding thermodynamics and presence of regulatory splicing motifs. All the designed ASOs were synthesized by GenePharma (China) as single-stranded 2'-O-methyl modified RNA bases linked with phosphorothioate backbone. Each ASO inducing specific exon-skipping events in GOLGA2 gene was designed to bind exonic splicing enhancer sequence to mask respective splicing motifs that were required for the proper splicing of the target exon. ASOs were transfected by DharmaFECT 1 transfection reagent (GE) or electroporated using the Nucleofector™ 2b Device (Lonza) into cells at a final concentration of 100–300 nM.

**Chromatin-associated RNA isolation and snRNA qRT-PCR.** Chromatin-associated RNA was isolated by a modification of the method developed by previous reports[37,66]. Briefly, cell pellets were resuspended in 20 mM HEPES (pH 7.5), 10 mM KCl, 250 mM sucrose, 5 mM MgCl₂, 1 mM EGTA, 1 mM PMSF, 1 μl/ml RNase inhibitor (Takara), 1× phosphatase inhibitor (Biotool) and 1× protease inhibitor cocktail (Calbiochem), and lysed by the addition of Digitonin (Sigma) to 200 μg/ml final concentration (10 min, 4 °C). Nuclei were pelleted by centrifugation (650 × *g*, 5 min) and, following re-suspension in a buffer containing 20 mM Tris-HCl (pH 7.5), 75 mM NaCl, 0.5 mM EGTA, 50% glycerol, 1 mM PMSF, 1 μl/ml RNase inhibitor (Takara), 1× phosphatase inhibitor (Biotool) and 1× protease inhibitor cocktail (Calbiochem), were extracted for 10 min at 4 °C by the addition of 10 volumes of a solution containing 20 mM HEPES (pH 7.6), 7.5 mM MgCl₂, 0.2 mM EGTA, 300 mM NaCl, 1 M urea and 1% NP-40. Pelleted nuclei were resuspended in Trizol reagent (Invitrogen) and RNA was isolated as recommended by the manufacturer. Following digestion with DNase I (Qiagen), equal amounts of RNA from each sample were reverse-transcribed. qRT-PCR reactions were performed using primers complementary to human or mouse snRNAs or 18S (control for data normalization), using the SYBR Green PCR Master Mix (Applied Biosystems, Foster City, CA). Primers are shown in Supplementary Table 4. Absence of contaminating genomic DNA was verified by the lack of amplified products for all sample/primer sets by inclusion of mock reverse-transcription reactions in which no enzyme was added.

**ASOs.** Novel ASOs were applied to bind to nascent transcripts of a target gene via Watson-Crick bonding to exert steric hindrance effect against splicing factors to modulate splicing. The ASOs were rationally designed for optimal efficiency in inducing splicing modulation. Briefly, ASO target sites were selected by a computational algorithm (ESE finder, human splicing finder) that accounted for co-

transcriptional binding accessibilities, binding thermodynamics and presence of regulatory splicing motifs. All the designed ASOs were synthesized by GenePharma (China) as single-stranded 2'-O-methyl modified RNA bases linked with phosphorothioate backbone. Each ASO inducing specific exon-skipping events in GOLGA2 gene was designed to bind exonic splicing enhancer sequence to mask respective splicing motifs that were required for the proper splicing of the target exon. ASOs were transfected by DharmaFECT 1 transfection reagent (GE) or electroporated using the Nucleofector™ 2b Device (Lonza) into cells at a final concentration of 100–300 nM.

**Electron microscopy**. Whole cells were fixed in 2% glutaraldehyde in PBS for 12 h at 4 °C, incubated in 1× PBS for 10 min 2 times to remove free glutaraldehyde, post-fixed at 1% osmium tetroxide in PBS for 2 h, then dehydrated through a gradual series of ethanol to 100%, and incubated in propylene oxide for 10 min 2 times, transferred into a mixture of 1:1 (v/v) resin/propylene oxide for 2 h, and transferred into a mixture of 2:1 (v/v) resin/propylene oxide for overnight. Samples were transferred to fresh resin and applied to filling embedding capsules for 48 h at 37 °C. Ultra-thin sections were cut using an ultramicrotome, and analyzed on a Philips CM-120 electron microscope at 80 kV.

**Endothelial recruitment assay**. A total of 50,000 cancer cells were seeded into 24-well plates approximately 24 h prior to the start of the assay. HUVECs were serum-starved in DMEM supplemented with 0.2% FBS for 20 h. Cancer cells were washed with PBS and 1 ml of 0.2% FBS DMEM was added to each well. Each well was then fitted with a Transwell Permeable Supports 6.5 mm Insert (Costar). In all, 80,000 HUVECs, resuspended in 0.2 ml of starvation media, were seeded into each insert and incubated at 37 °C for 18 h.

**Mouse studies**. DU145 cells ($3 \times 10^6$ each mouse), SW620 cells ($3 \times 10^6$ each mouse) or U251 cells ($2.5 \times 10^6$ each mouse) were inoculated subcutaneously into nude mice (Shanghai Laboratory of Animal Center, Chinese Academy of Sciences) and tumor volume was monitored. Animal care and experiments were performed in strict accordance with the "Guide for the Care and Use of Laboratory Animals" and the "Principles for the Utilization and Care of Vertebrate Animals" and were approved by the committee for humane treatment of animals at Shanghai Jiao Tong University School of Medicine.

**Apoptosis assay**. Cells were initially seeded into 12-well plates with $1 \times 10^5$ per well and incubated with DMSO, BFA and GCA with proper concentrations and times. Annexin-V assay was performed on flow cytometry (BD Biosciences) according to the manufacturer's instruction provided by the ApoAlert Annexin-V kit (BD Clontech). Here, cell death was determined by positive staining of Annexin-V and propidium iodide (PI) alone, or both.

**Data availability**. All sequencing data that support the findings of this study have been deposited in the NCBI Sequence Read Archive and are accessible through the accession number SRP147969. The mass spectrometry proteomics data have been deposited to the ProteomeXchange Consortium via the PRIDE partner repository with the dataset identifier PXD009681 and 10.6019/PXD009681[67]. Microarray data have been deposited in the Protein Microarray Database and are accessible through the accession number PMDE231. All other relevant data are available within the article and its Supplementary Information Files, or from the corresponding author on request.

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

## Acknowledgements

We thank Professor Hongbing Zhang, Ian C. Eperon, Adrian R. Krainer, Ping-Jin Gao and Jiahuai Han for kindly providing materials as described in the Methods section. This work was supported by National Natural Science Foundation (81230048; 81430061; 81502001) and its innovative group support (No.81721004); National Key Research Program of China (NO2015CB910403, NO2013CB910903); Natural Science Foundation from Science and Technology committee of Shanghai (15ZR1426600; 16ZR1449900) as well as the Fundamental Research Funds for the Central Universities.

## Author contributions

S.-M.S. designed and performed most experiments. Y.J. performed bioinformatics analysis. C.Z., S.-S.D., Z.X. and M.-K.G. conducted partial experiments. L.X. performed LC−MS/MS.Y.Y., M.G. and S.Y. contributed to results discussion. J.-K.C., J.-L.L and J.-X.Y. provided constructive comments and discussion. G.-Q.C. and S.-M.S. designed and supervised the entire project and prepared the manuscript.

## Additional information

**Competing interests:** The authors declare no competing interests.

