## [Peer Review File · Nature Communications]

Reviewer #1 (Remarks to the Author):

The paper by Shen et al. is potentially interesting but contains two messages, which should be presented in two different papers. The first part of the paper about alternative splicing is more or less understandable. The second part about the Golgi is not well defined. The images are too small and it is very difficult to see what is presented there. Electron microscopic images represent a mess. I cannot see alterations of the Golgi there. Immune fluorescence images are also too small and do not show the phenomena described. It is difficult to evaluate the quality of EM images. They do not illustrate the messages of the paper and without quantification are useless. Letters in graphs and images cannot be read at normal magnification (100%). In Fig. 5I there is no proof of the message and it is invisible. Figure 5G is also very bad. Fig. 7 contains 130 pages. Fig. S4 is very bad. What is the meaning of "exofacial VSVG" in image caption?

Reviewer #2 (Remarks to the Author):

Title:

Nuclear PTEN safeguards pre-mRNA splicing to link Golgi extension and secretion for its tumor suppressive role

Recommendation:

Major revision

Summary:

In this study, Shen et al. report that nuclear PTEN interacts with the splicing machinery, spliceosome, to regulate its assembly and subsequent pre-mRNA splicing in a phosphatase-independent manner. They further demonstrate that increased growth of PTEN-knockdown cancer cells is dependent on enhanced Golgi extension and secretion, which is mediated by a GOLGA2 protein isoform, generated from a PTEN-loss induced exon-skipping mechanism. This, in turn, creates vulnerability to Golgi secretion inhibition for PTEN-deficient cancer. This is an interesting study and adds to our new understanding of nuclear PTEN function in the tumor suppression. However, the study should be strengthened by additional data to corroborate PTEN scaffold function in the regulation of pre-mRNA splicing. Importantly, critical controls should be added in the experimental plan. Specific comments that should be addressed to improve the quality of the data are:

Major points:

1) The PTEN scaffold function in the regulation of alternative splicing needs to be further corroborated. Does PTEN C124S affect alternative splicing in a similar manner as WT PTEN in 293T cells, through RNA-seq?

2) Additionally, does restoration of WT or C124S PTEN result in similar tumor suppression in cancer cells with high exon-skipped GOLGA2 isoform? These experiments are of critical importance to substantiate the authors' claim.

3) Which functional domain of PTEN interacts with U2AF2?

4) Since RT-PCR is not quantitative, q-PCR should be employed instead to quantify the ratio of alternative splicing for genes of interest throughout the study.

5) Most of the methods are not provided. Please include the relevant methods in the main text.

Minor points:

1) On Page 7, line 145, the authors should describe the results according to their respective figure order.

2) Fig. 2A and 2B are essentially the same and therefore redundant. This should be corrected.

3) On Page 36, line 848, "21" should be "20".

Reviewer #3 (Remarks to the Author):

In this study, Shen and co-authors investigate the role of the tumour suppressor PTEN in the regulation of alternative splicing (AS).

For this the authors first use splicing minigenes to assess the role of PTEN in constitutive and alternative splicing. This was followed by the analysis of the global impact of PTEN on AS, using RNA-

seq of 293T cells depleted of PTEN. This led to the identification of a large number of PTEN-regulated AS events, out of which 262 were common to PTEN knock-down by two different shRNAs. The authors went on to show that the PTEN-regulated AS events are generally linked to cancer progression, as seen with analysis of TCGA tumor collections. Some of these AS events are associated with a significant worse outcome in cancer patients.

The authors next show that nuclear PTEN associates with the splicing machinery in the nucleus and regulate its assembly and function. In particular, the authors focused on the interaction of PTEN with the RS domain of the the splicing factor U2AF2. Next, Shen and colleagues focused on a particular AS event, the increased skipping of exon 2b in the GOLGA2 pre-mRNA, which is observed upon PTEN depletion. This regulated event gives rise to a protein isoform, GOLGA2S that promotes Golgi extension and secretion and contributes to PTEN knockdown-induced tumorigenesis. Finally, the authors show that PTEN depletion sensitizes cancer cells to secretion inhibitors.

Overall, this is an interesting study that has been well designed and presents clear data. The manuscript is well-written and the conclusions are fully supported by clear data. In principle, this study could be suitable for Nature communications. The manuscript would benefit from the following revisions.

Specific comments

- Results on Fig. 1A with the AS of the tropomyosin minigene are not very convincing. The effects seem very modest. The quantitations seems to indicate a larger effect to what is seen in the gel. A similar observation applies to Fig. 1B. These minigenes do not seem particularly responsive to levels of PTEN. Have the authors tried other widely used AS minigenes, such as Fibronectin EDA (Kornblihtt lab) or the E1A adenovirus minigene (Kraimer and Cáceres labs)?

- Concerning the issue described above, the authors need to use minigenes that show a clear response to PTEN. Other solution would be to completely get rid of the Minigene systems and assay for PTEN effect in splicing in a genome-wide manner as they do on Fig. 1E.

- On Fig. 4, the authors show that the interaction of PTEN interacts with the RS domain of U2AF2. Have the authors tested whether phosphorylation of this domain affects this interaction? This could be done with phosphomimetic mutants of U2AF2

Minor

- Ref 8 is incorrect, since it is referring to a Corrigendum of an actual paper, which is the one that should be cited

Reviewer #4 (Remarks to the Author):

The manuscript by Shen et al. reports the intriguing observation that nuclear PTEN can interact with the spliceosome (by directly binding U2AF2) and drives the alternative splicing of a number of transcripts, including that of GOLGA2, KIF21A and CD2CD5. The authors hypothesize that this novel activity of PTEN, which is independent of its phosphatase activity, may be relevant for its oncosuppressor role and select GOLGA2, encoding the Golgi matrix protein GM130, to perform what they define an “in depth functional analysis”.

The authors find that Hek293 express a previously undescribed isoform of GM130 carrying an additional exon (2b) and that the depletion of PTEN induces the expression of an isoform devoid of exon2b, GOLGA2S.

The authors report that the loss of PTEN induces an enlargement of the Golgi complex, evaluated by IF and EM, and an increased rate of trafficking along the secretory pathway and that these phenotypes can be reversed by specifically depleting GOLGA2S with shRNA targeting the exon2-3 junction. Finally, they find that the loss of PTEN induces a higher rate of xenografts that can be reversed by KD of GOLGA2S.

Although the observation of the alternative splicing induced by PTEN loss is in principle of interest, the mechanistic analysis of the consequences of this is at a very preliminary stage thus leaving the conclusions of the manuscript largely unjustified.

Indeed, there is a complete lack of analysis of the two GOLGA2 isoforms in terms of protein products with respect to their MW, stability, localization, interactors and function. This analysis is absolutely necessary since the data presented by the authors would suggest that GOLGA2S exerts a role that appears to be opposite to that so far described for GM130 (for instance, GOLGA2S depletion induces lengthening of Golgi cisternae whereas GM130 depletion induces fragmentation of the Golgi ribbon).

Specific points

The analysis of VSVG transport along the secretory pathway is not convincing and higher resolution analysis and images are needed.

GOLGA2S appears to be present also in PTEN-expressing cells and the imbalance in alternative splicing induced by PTEN loss is variable in different cell lines.

Detailed response to reviewers' comments for manuscript NCOMMS-17-24294

Dear all referees:

We appreciate your positive, professional and serious evaluations on our manuscript NCOMMS-17-24294. In the past several months, we have seriously revised the manuscript according to your constructive comments with the supplement of some new experiments presented in the current Fig. 1a, Fig 6a/b, Supplementary Fig. 2C/D, Supplementary Fig. 3F-I, Supplementary Fig. 5H-K, and Supplementary Fig. 6A/B. Totally, we have done our best to address all of your comments, as mentioned in the following point-by-point responses (**black** words are your comments and **blue** words are our responses). We appreciate that you can find that the version is acceptable for publication in Nat Commun.

Kind regards,

Guo-Qiang Chen, M.D., Ph.D.

Academician, Chinese Academy of Science
President, Shanghai Jiao Tong University School of Medicine
Vice president, Shanghai Jiao Tong University
Professor and Director, Key Laboratory of Cell Differentiation and Apoptosis of Chinese Ministry of Education, Shanghai Jiao Tong University School of Medicine
Address: 280, Chong-Qing South Road, Shanghai 200025, China
Email: chengq@shsmu.edu.cn; gqchen@sibs.ac.cn

Responses for Reviewer #1:

The paper by Shen et al. is potentially interesting but contains two messages, which should be presented in two different papers. The first part of the paper about alternative splicing is more or less understandable. The second part about the Golgi is not well defined.

RESPONSE: Thanks the reviewer for his/her interest in the paper. Actually, this work

mainly report that nuclear PTEN can interact with the spliceosome and drives the alternative splicing of a number of transcripts, including that of GOLGA2. This novel activity of PTEN, which is independent of its phosphatase activity, may be relevant for its oncosuppressor role. We selected GOLGA2 to perform an “in depth functional analysis”. Therefore, we do not think that these messages should be presented in two different papers, although the manuscript carries many original data.

The images are two small and it is very difficult to see what is presented there.

Electron microscopic images represent a mess. I cannot see alterations of the Golgi there. Immune fluorescence images are also too small and do not show the phenomena described. It is difficult to evaluate the quality of EM images. They do not illustrate the messages of the paper and without quantification are useless. Letters in graphs and images cannot be read at normal magnification (100%). In Fig. 5I there is no proof of the message and it is invisible. Figure 5G is also very bad. Fig. 7 contains 130 pages. Fig. S4 is very bad.

RESPONSE: Sorry. We do not know what happen. Actually we could clearly show all data on our computer when we submitted. It appeared that all other three reviewers can also clearly read the manuscript and all the figures. Of course, there are too many panels in one figure, making it more difficult to be shown in print version of A4 paper. We hope you see the figures on a desktop computer with the images magnified. We also feel surprised that you find Fig. 7 contains 130 pages. Maybe our previous supplementary tables in excel format were transformed to PDF and caused this problem. In this verison, we correctly put these data as supplementary dataset as excel files. Thanks.

What is the meaning of "exofacial VSVG" in image subscription?

RESPONSE: As addressed in the text and in the corresponding Figure legend (current Figure 6a), “exofacial VSVG” is the portion of VSVG that can be detected by antibody against the extracellular domain of VSVG in unpermeabilized cells (see also ref#53), which allows unambiguous detection of VSVG on the plasma membrane. Fig. 6a showed that wild-type MEF cells had little exofacial-VSVG on plasma membrane, which became more significant in PTEN^{-/-} MEF cells, suggesting that PTEN depletion robustly increased trafficking of ts045-VSVG-EGFP to the plasma membrane in MEF cells (Fig. 6a). This is also true in DU145 cells with knockdown of PTEN (Fig. 6b). For detail, please see lines 424-433 of page 18 and also the legend of Fig.6a, b.

Responses for Reviewer #2:

Summary: In this study, Shen et al. report that nuclear PTEN interacts with the splicing machinery, spliceosome, to regulate its assembly and subsequent pre-mRNA splicing in a phosphatase-independent manner. They further demonstrate that increased growth of PTEN-knockdown cancer cells is dependent on enhanced Golgi extension and secretion, which is mediated by a GOLGA2 protein isoform, generating from a PTEN-loss induced exon-skipping mechanism. This, in turn, creates vulnerability to Golgi secretion inhibition for PTEN-deficient cancer. This is an interesting study and adds to our new understanding of nuclear PTEN function in the tumor suppression. However, the study should be strengthened by additional data to corroborate PTEN scaffold function in the regulation of pre-mRNA splicing. Importantly, critical controls should be added in the experimental plan.

RESPONSE: Thanks for your interest and positive and professional evaluations on

this study. We have revised the manuscript according to your comments, which should greatly strengthen the work. Especially, the scaffold function of PTEN in the regulation of pre-mRNA splicing has been further evaluated.

Major point 1: The PTEN scaffold function in the regulation of alternative splicing needs to be further corroborated. Does PTEN C124S affect alternative splicing in a similar manner as WT PTEN in 293T cells, through RNA-seq?

RESPONSE: Thanks. In the previous submission, we demonstrated that re-expression of either wild-type PTEN or PTEN^{C124S} in PTEN^{-/-} MEF cells reversed the effect of PTEN depletion on pTN24 splicing (current Fig. 1b). Also, PTEN phosphatase inhibitors or AKT inhibitors failed to recapitulate or reverse the effect of PTEN depletion on pTN24 splicing (current Supplementary Fig. 2A, B). All of these results proposed that PTEN regulates alternative splicing in its phosphatase-independent manner. According to the comment so as to consolidate the PTEN scaffold function in the regulation of alternative splicing, we stably transduced PTEN^{WT} or PTEN^{C124S}-expressing plasmids, both of which are synonymous mutants resistant to shPTEN#1, together with an empty vector into 293T-shPTEN#1 cells, followed by RNA-seq and analysis for ASEs. Our results revealed that the expression of PTEN^{WT} and PTEN^{C124S} effectively reversed the ASEs induced by shPTEN#1 to the similar degree, in which PTEN^{WT} reversed 83.6% (877 out of 1048, Supplementary Data 2) and PTEN^{C124S} reversed 81.7% (856 out of 1048, Supplementary Data 2) of the shPTEN#1 induced ASEs (Supplementary Fig. 2C, D). These results further support that the phosphatase-independent activities of PTEN contribute to its regulation on alternative splicing. The related descriptions can be seen in lines 189-197 of page 9.

Major point 2: Additionally, does restoration of WT or C124S PTEN result in similar tumor suppression in cancer cells with high exon-skipped GOLGA2 isoform?

These experiments are of critical importance to substantiate the authors' claim.

RESPONSE: A good comment. In the revised version, we transduced PTEN^{WT} and PTEN^{C124S}-expressing plasmids together with an empty vector by lenti-virus into PTEN-deficient U251 cells and tumorigenic capacities of these cells were monitored. Here, the U251 cell line was used because we have characterized U251 cells with high GOLGA2 exon 2b skipping and re-expression of PTEN^{WT} and PTEN^{C124S} in U251 cells decreased GOLGA2 exon-2b skipping to similar levels (current Supplementary Fig. 1C). Our results showed that, indeed, PTEN^{WT} and PTEN^{C124S} suppressed tumor growth with similar strengths in U251 cells (current Supplementary Fig. 6). The related descriptions have been added in the text (lines 451-456 of page 19).

Major point 3: Which functional domain of PTEN interacts with U2AF2?

RESPONSE: In the revised version, we mapped the functional domain of PTEN that interacts with U2AF2 to a fragment compassing amino acids 150-286 (current Supplementary Fig. 4G-I). This fragment compasses the C-terminal part of the phosphatase domain and the N-terminal part of the C2 domain of PTEN. The related descriptions have been given in the text (lines 310-312 of page 13). It should be pointed that we could not further refine the exact amino acids, because shorter fragments than amino acids 150-286 and PTEN with the fragment deletion could not be successfully expressed.

Major point 4: Since RT-PCR is not quantitative, q-PCR should be employed instead to quantify the ratio of alternative splicing for genes of interest throughout the study.

RESPONSE: Indeed, we tried both strategies to show our results in the beginning of our project. However, we found that RT-PCR rather than q-PCR has been used to show alternative splicing than in most literatures. We also finally chose RT-PCR for this, at least due to the following reasons. We hope the reviewer can agree with us. (1) Primer specificity. RT-PCR uses the same pair of primers to amplify both splicing products whose difference in sizes allowing for separation on the gel, while q-PCR uses two pairs of primers to amplify their respective splicing product. One of the primers that used to amplify the short product in q-PCR must be designed at the junction site, which occasionally raises issues such as non-specific amplification that must be addressed with extra efforts. (2) Principally, alternative splicing results in the change of proportions of two isoforms. Amplification with the same pair of primers in RT-PCR enables the comparison of the two isoforms. However, technically, due to the differences in amplification efficiency between primers, q-PCR products can't be directly compared to calculate the proportions of two isoforms. (3) There are special occasions such as the decreased levels of both isoforms due to the decreased gene expression, but at the same time alternative splicing takes place to change the proportion of the two isoforms. RT-PCR can detect both alterations, while alternative splicing might be overlooked by q-PCR under this circumstance. (4) Sometimes, alternative splicing between two exons generates more than two products, which can only be manifested by RT-PCR. In conclusion, we chose RT-PCR for these reasons, although we admit the advantage of q-PCR to quantitatively monitor the change of a certain isoform in an alternative splicing event.

Major point 5: Most of the methods are not provided. Please include the relevant

methods in the main text.

RESPONSE: Sorry, we provided methods in Supplementary information in the previous submission. In the revised version, we have put the Methods section into the text.

Minor point 1: On Page 7, line 145, the authors should describe the results according to their respective figure order.

RESPONSE: Thanks. We have modified this part in the revised version.

Minor point 2: Fig. 2A and 2B are essentially the same and therefore redundant. This should be corrected.

RESPONSE: We deleted the previous Figure 2B.

Minor point 3: On Page 36, line 848, “21” should be “20”.

RESPONSE: Sorry for this mis-spelling. We have corrected it in the revised version.

Responses for Reviewer #3:

Summary: In this study, Shen and co-authors investigate the role of the tumour suppressor PTEN in the regulation of alternative splicing (AS). For this the authors first use splicing minigenes to assess the role of PTEN in constitutive and alternative splicing. This was followed by the analysis of the global impact of PTEN on AS, using RNA-seq of 293T cells depleted of PTEN. This led to the identification of a large number of PTEN-regulated AS events, out of which 262 were common to PTEN knock-down by two different shRNAs. The authors went on to show that the PTEN-regulated AS events are generally linked to cancer progression, as seen with analysis of TCGA tumor collections. Some of these AS events are associated with a significant worse outcome in cancer patients. The authors next show that nuclear PTEN associates with the

splicing machinery in the nucleus and regulate its assembly and function. In particular, the authors focused on the interaction of PTEN with the RS domain of the splicing factor U2AF2. Next, Shen and colleagues focused on a particular AS event, the increased skipping of exon 2b in the GOLGA2 pre-mRNA, which is observed upon PTEN depletion. This regulated event gives rise to a protein isoform, GOLGA2^S that promotes Golgi extension and secretion and contributes to PTEN knockdown-induced tumorigenesis. Finally, the authors show that PTEN depletion sensitizes cancer cells to secretion inhibitors. Overall, this is an interesting study that has been well designed and presents clear data. The manuscript is well-written and the conclusions are fully supported by clear data. In principle, this study could be suitable for Nature communications.

RESPONSE: Thanks for your good summary and enthusiasm on this study.

Comment: Results on Fig. 1A with the AS of the tropomyosin minigene are not very convincing. The effects seem very modest. The quantitations seem to indicate a larger effect to what is seen in the gel. A similar observation applies to Fig. 1B. These minigenes do not seem particularly responsive to levels of PTEN. Have the authors tried other widely used AS minigenes, such as Fibronectin EDA (Kornblihtt lab) or the E1A adenovirus minigene (Krainer and Caceres labs)?

Concerning the issue described above, the authors need to use minigenes that show a clear response to PTEN. Other solution would be to completely get rid of the Minigene systems and assay for PTEN effect in splicing in a genome-wide manner as they do on Fig. 1E.

RESPONSE: According to this and following concerns, we got rid of results from the

tropomyosin minigene (previous Fig. 1A) and also results by pTN24 in 293T cells (previous Fig. 1C). We keep the results obtained with pTN24 in MEF cells (current Fig. 1b). Also, we got the E1A adenovirus minigene from professor Krainer, and experiments with this minigene still supported the role of PTEN on alternative splicing (see lines 118-123 of page 6), as shown in current Fig. 1a.

Comment: On Fig. 4, the authors show that the interaction of PTEN interacts with the RS domain of U2AF2. Have the authors tested whether phosphorylation of this domain affects this interaction? This could be done with phosphomimetic mutants of U2AF2.

RESPONSE: Good comment! Indeed, RS domains are normally subjected to phosphorylation in serine-arginine proteins [PMID: 21205204]. To our greater understanding, however, phosphorylated sites in the RS domain of U2AF2 were not identified in previous investigations. Therefore, we analyzed the RS domain of U2AF2 with Scansite tool and found that seven serines (S30/32/34/41/43/55/62) are its potential phosphorylation sites. We thus constructed a phosphomimetic mutant of U2AF2 RS domain by substituting all these seven serines (S) with aspartic acids (D), and performed co-immunoprecipitation. The result showed that while PTEN strongly bound wild-type U2AF2 RS domain, it failed to bind the phosphomimetic mutant, suggesting that phosphorylation of U2AF2 in the RS domain might disrupt the interaction between PTEN and U2AF2 (see lines 301-309 of page 13 and current Supplementary Fig. 4F).

Minor point: Ref 8 is incorrect, since it is referring to a Corrigendum of an actual paper, which is the one that should be cited.

RESPONSE: Thanks. We have corrected it.

Response for Reviewer #4:

Comment: The manuscript by Shen et al. reports the intriguing observation that nuclear PTEN can interact with the spliceosome (by directly binding U2AF2) and drives the alternative splicing of a number of transcripts, including that of GOLGA2, KIF21A and CD2CD5. The authors hypothesize that this novel activity of PTEN, which is independent of its phosphatase activity, may be relevant for its oncosuppressor role and select GOLGA2, encoding the Golgi matrix protein GM130, to perform what they define an “in depth functional analysis”. The authors find that Hek293 express a previously undescribed isoform of GM130 carrying an additional exon (2b) and that the depletion of PTEN induces the expression of an isoform devoid of exon2b, GOLGA2^S. The authors report that the loss of PTEN induces an enlargement of the Golgi complex, evaluated by IF and EM, and an increased rate of trafficking along the secretory pathway and that these phenotypes can be reversed by specifically depleting GOLGA2^S with shRNA targeting the exon2-3 junction. Finally, they find that the loss of PTEN induces a higher rate of xenografts that can be reversed by KD of GOLGA2^S.

Although the observation of the alternative splicing induced by PTEN loss is in principle of interest, the mechanistic analysis of the consequences of this is at a very preliminary stage thus leaving the conclusions of the manuscript largely unjustified.

Indeed, there is a complete lack of analysis of the two GOLGA2 isoforms in terms of protein products with respect to their MW, stability, localization, interactors

and function. This analysis is absolutely necessary since the data presented by the authors would suggest that GOLGA2^S exerts a role that appears to be opposite to that so far described for GM130 (for instance, GOLGA2^S depletion induces lengthening of Golgi cisternae whereas GM130 depletion induces fragmentation of the Golgi ribbon).

RESPONSE: Thanks for your good comments on this study. As described in the text, our immunofluorescent staining with anti-GM130 antibody showed that DU145 (Fig. 5d) and NCM-460 (Supplementary Fig. 5C) cells expressing ASOs which induced increased GOLGA2^S (Figure 5c), exhibited more extended distribution of perinuclear Golgi staining. Moreover, specific knockdown of the GOLGA2^S (Fig. 5h) almost completely reversed the phenotypic effect of PTEN depletion on Golgi extension (Fig. 5i), supporting that the ability of PTEN depletion to promote Golgi extension was dependent on the expression of GOLGA2^S. These results should express that GOLGA2^S exerts the same role to that described for GM130.

In this revised version, we performed some analysis on the two GOLGA2 isoforms in terms of protein products in the current Supplementary Fig. 5H-K. The related following description can be seen in lines 392-419 of pages 17-18: The mRNA sequence of GOLGA2^S matches to that of canonical GOLGA2, a protein coding gene whose protein product is GM130. Theoretically, exon 2b (81 bp) inclusion will not result in frameshift or premature termination, meaning that GOLGA2^L should encode a protein product with extra 27 amino acids in the N-terminus of GM130 and with ~3kD difference in molecular weight compared to GM130 (Supplementary Fig. 5H). Unexpectedly, an antibody recognizing the C-terminus of GM130, which should detect both

GM130 and GOLGA2^L-encoding protein, only detected a single band in several cell lines subjected to Western blot (Supplementary Fig. 5I). Interestingly, the level of this band was increased by PTEN knockdown in all three cell lines tested (Supplementary Fig. 5I). There might be possibilities that the protein products of the two isoforms were not well separated on gel or only one of the isoforms resulted in protein production. To find out which was the case, we tried to decide the identity of this band. We transfected DU145 and 293T cells with shRNAs respectively targeting GOLGA2^S or GOLGA2^L. As shown in Supplementary Fig. 5J, both shRNAs efficiently knocked down their intended targets without affecting each other. However, only shRNA targeting GOLGA2^S significantly reduced the band, while knockdown of GOLGA2^L did not affect the band (Supplementary Fig. 5J), indicating that GM130 encoded by GOLGA2^S constituted the majority of the protein products. Thus, PTEN knockdown increased GM130 by promoting GOLGA2 exon 2b skipping. We then asked whether GOLGA2^L encoded a protein product with very low stability. For this end, cells were treated with MG132, but no accumulated band was observed (Supplementary Fig. 5K). Thus, we conclude that canonical GM130 encoded by GOLGA2^S is the only detectable protein product of GOLGA2 isoforms. However, we could not rule out the possibility that GOLGA2^L might act as a non-coding RNA. Actually, several protein-coding genes have also been reported to generate long non-coding RNA (lncRNA) isoforms by alternative splicing to play a distinct role (ref#52). Whether GOLGA2^L also acts as a non-coding RNA will definitely be interesting for us to explore in our future studies.

Specific points: The analysis of VSVG transport along the secretory pathway is not

convincing and higher resolution analysis and images are needed.

RESPONSE: We repeated the related experiments and obtained same results.

Considering this reviewer's concern, we used Leica TCS SP8 to capture images with higher resolutions (current Fig. 6a, b), which should be clearer than our previous images. Of course, we also revised the related description so as to be understood clearly for this experiment in lines 423-438 of page 18-19.

Comment 3: GOLGA2^S appears to be present also in PTEN-expressing cells and the imbalance in alternative splicing induced by PTEN loss is variable in different cell lines.

RESPONSE: Yes. In this version, we pointed out that "PTEN was not the sole decisive factor of GOLGA2 exon 2b splicing because GOLGA2^S appears to be present also in PTEN-expressing cells, and the imbalance in alternative splicing induced by PTEN loss is variable in different cell lines" (see lines 361-363 of page 15-16). Indeed, the regulation of GOLGA2^S by PTEN is not an all-or-none effect. We speculate that the presence of PTEN makes the splicing of exon-2b more efficient, and loss of PTEN leads to reduced splicing efficiency instead of completely abolishing exon-2b splicing. This speculation is compatible with our observations in other parts of the manuscript concerning the regulatory role of PTEN. Additionally, the imbalance in alternative splicing induced by PTEN loss is variable in different cell lines. We reason that there are two major factors that decide the extent of the effect induced by PTEN loss: the basal level of exon-2b skipping, and the extent of PTEN's role before knockdown. These two factors varies in different cell lines.

REVIEWERS' COMMENTS:

Reviewer #2, Expertise: PTEN, cancer (Remarks to the Author):

Recommendation:

Accept

Summary:

The authors have adequately address my questions and the revised manuscript has been significantly strengthened.

Reviewer #3, Expertise: pre mRNA splicing (Remarks to the Author):

The authors have performed a thorough revision and have either addressed or attempted to address the concerns raised during the first round of reviews. It is my view that they have done a good job and that the revised manuscript is improved. I think that this paper is now acceptable for publication in Nature communications.

Reviewer #4, Expertise: Golgi, secretory pathway (Remarks to the Author):

In the revised manuscript the authors have partially addressed the questions concerning the analysis of the GOLGA2 isoforms in terms of protein expression by analyzing the GM130 protein by Western blot in cells depleted of PTEN (Fig. 5i) and in cells transfected with specific shRNAs against GOLGA2S and GOLGA2L (Fig. S5j and k).

As the authors could detect only one GM130 protein band (irrespective of the presence or absence of the GOLGA2L transcript), they conclude that the GOLGA2L transcript does not give rise to any protein product. Ideally, this conclusion should be reinforced by developing an Ab against the 28 amino acid insert of GOLGA2L or by sequencing the GM130 in immunoprecipitates obtained with the available anti-GM130 Ab.

The authors observe that the level of GM130 protein is higher in PTEN-depleted cells (Fig. S5i) and thus hypothesize that the lowering of GOLGA2L transcript in PTEN-depleted cells favors the production of the protein product of the GOLGA2S transcript.

The authors then assess the effect of selectively inhibiting (via specific shRNAs) the transcription of GOLGA2S or GOLGA2L on the levels of the reciprocal mRNA and on the GM130 protein.

With regard to the levels of mRNA, consistent with the finding in PTEN-depleted cells, they find that depletion of the GOLGA2L transcript increases the levels of the GOLGA2S transcript. However, when they analyze the protein levels, they get results that are difficult to interpret.

Firstly, the increased levels of the GOLGA2S transcript (under conditions of treatment with shRNAs against GOLGA2L) do not result in an increase in GM130 levels (thus not mimicking the effect of PTEN depletion). Secondly, in Fig. S5k they observe an increase of GM130 in 293T cells treated with shRNAs against GOLGA2S.

Thus, the new set of data does not really help in reaching conclusive information about the role of the GOLGA2L isoform and how this isoform may compete and/or interact with GOLGA2S. The authors state that GOLGA2S is the main isoform present in the different cell lines tested and that its levels increase in the absence of PTEN. However, considering the lack of clear effects on the level of GM130 protein induced by the specific and complete depletion of GOLGA2L, it looks unlikely that the marked increase of GM130 protein levels in PTEN-KD cells may be due only to the imbalance of alternative splicing of the GOLGA2 transcript towards the S isoform since, in most of the cell lines used (MEF, SF188, RKO and HeLa), the L isoform is still predominant.

In regards of the VSVG trafficking assay, the authors have provided better images, but they should also perform a quantitative analysis.

Detailed response to reviewers' comments for manuscript NCOMMS-17-24294A

Dear reviewer #4:

We appreciate your professional and serious evaluations on our manuscript NCOMMS-17-24294A. We have done our best to address your comments, as mentioned in the following point-by-point responses (**black** words are your comments and **blue** words are our responses). We sincerely hope that you can find that the version is now acceptable for publication in Nature Communications.

Kind regards,

Guo-Qiang Chen, M.D., Ph.D.

Responses for Reviewer #4:

In the revised manuscript the authors have partially addressed the questions concerning the analysis of the GOLGA2 isoforms in terms of protein expression by analyzing the GM130 protein by Western blot in cells depleted of PTEN (Fig. S5i) and in cells transfected with specific shRNAs against GOLGA2S and GOLGA2L (Fig. S5j and k). As the authors could detect only one GM130 protein band (irrespective of the presence or absence of the GOLGA2L transcript), they conclude that the GOLGA2L transcript does not give rise to any protein product. Ideally, this conclusion should be reinforced by developing an Ab against the 28 amino acid insert of GOLGA2L or by sequencing the GM130 in immunoprecipitates obtained with the available anti-GM130 Ab.

RESPONSE: We agree with this concern. Actually, as the reviewer has recommended, we have immunoprecipitated the endogenous GM130 with the available anti-GM130 Ab and sequenced it with mass spectrometry, but found no trace of peptides matching the 27 amino acid insert of GOLGA2^L. We also constructed plasmids encompassing the whole CDS of GOLGA2^S and GOLGA2^L and transfected them into 293T cells. RT-PCR detected forced expression of both mRNAs, and quantitative real-time PCR using primers quantified the expression efficiencies to be ~200 times compared to endogenous mRNAs. However, western blot only detected the product of

GOLGA2^S, which was the same molecular weight of endogenous GM130. An ideally higher molecular weight band that corresponds to the product of GOLGA2^L was not observed. We did not choose to show these results in the current manuscript. However, we have made considerable revisions to this part in the current version of our manuscript, and in the last paragraph of discussion, we pointed out “Whether GOLGA2^L transcript give rises to any protein product remained to be further confirmed by developing an antibody against the 28 amino acid insert of GOLGA2^L. However, we could not rule out the possibility that GOLGA2L might act as a non-coding RNA. Actually, several protein-coding genes have also been reported to generate long non-coding RNA (lncRNA) isoforms by AS to play a distinct role⁵⁷. Whether GOLGA2^L also acts as a non-coding RNA will definitely be interesting for us to explore in our future studies”. We would continue these interesting works in the future.

The authors observe that the level of GM130 protein is higher in PTEN-depleted cells (Fig. S5i) and thus hypothesize that the lowering of GOLGA2L transcript in PTEN-depleted cells favors the production of the protein product of the GOLGA2S transcript. The authors then assess the effect of selectively inhibiting (via specific shRNAs) the transcription of GOLGA2S or GOLGA2L on the levels of the reciprocal mRNA and on the GM130 protein. With regard to the levels of mRNA, consistent with the finding in PTEN-depleted cells, they find that depletion of the GOLGA2L transcript increases the levels of the GOLGA2S transcript. However, when they analyze the protein levels, they get results that are difficult to interpret. Firstly, the increased levels of the GOLGA2S transcript (under conditions of treatment with shRNAs against GOLGA2L) do not result in an increase in GM130 levels (thus not mimicking the effect of PTEN depletion). Secondly, in Fig. S5k they observe an increase of GM130 in 293T cells treated with shRNAs against GOLGA2S. Thus, the new set of data does not really help in reaching conclusive information about the role of the GOLGA2L isoform and how this isoform may compete and/or interact with GOLGA2S.

The authors state that GOLGA2S is the main isoform present in the different cell lines tested and that its levels increase in the absence of PTEN. However,

considering the lack of clear effects on the level of GM130 protein induced by the specific and complete depletion of GOLGA2L, it looks unlikely that the marked increase of GM130 protein levels in PTEN-KD cells may be due only to the imbalance of alternative splicing of the GOLGA2 transcript towards the S isoform since, in most of the cell lines used (MEF, SF188, RKO and HeLa), the L isoform is still predominant.

RESPONSE: We have modified the description in the related results section to avoid premature assumptions. Besides, we have quantified the data on GM130 expression upon depletion of GLOGA2 isoforms in the current Fig. S8C. Sorry, we made a mistake with previous Fig. S5K, in which sample orders were not actually the same for DU145 and 293T cells. This mistake made you be confused. We have corrected it in the revised Fig S8D. We are sorry for the confusion caused by this mistake.

In regards of the VSVG trafficking assay, the authors have provided better images, but they should also perform a quantitative analysis.

RESPONSE: We have performed quantitative analysis for the VSVG trafficking assay (see current Figure 6b, d).